# Resilience of genetic diversity in forest trees over the Quaternary

Pascal Milesi [1,2,23] ✉, Chedly Kastally [3,4,23], Benjamin Dauphin [5,23], Sandra Cervantes [6,7,23], Francesca Bagnoli [8], Katharina B. Budde[9,10], Stephen Cavers [11], Bruno Fady[12], Patricia Faivre-Rampant [13], Santiago C. González-Martínez [14], Delphine Grivet [15], Felix Gugerli [5], Véronique Jorge [16], Isabelle Lesur Kupin[14,17], Dario I. Ojeda[18], Sanna Olsson [15], Lars Opgenoorth [5,19], Sara Pinosio [8,20], Christophe Plomion[14], Christian Rellstab [5], Odile Rogier [16], Simone Scalabrin [21], Ivan Scotti[12], Giovanni G. Vendramin [8], Marjana Westergren [22], Martin Lascoux [1,2,23] ✉ & Tanja Pyhäjärvi [3,4,23] ✉ On behalf of the GenTree Consortium*

The effect of past environmental changes on the demography and genetic diversity of natural populations remains a contentious issue and has rarely been investigated across multiple, phylogenetically distant species. Here, we perform comparative population genomic analyses and demographic inferences for seven widely distributed and ecologically contrasting European forest tree species based on concerted sampling of 164 populations across their natural ranges. For all seven species, the effective population size, $N_e$, increased or remained stable over many glacial cycles and up to 15 million years in the most extreme cases. Surprisingly, the drastic environmental changes associated with the Pleistocene glacial cycles have had little impact on the level of genetic diversity of dominant forest tree species, despite major shifts in their geographic ranges. Based on their trajectories of $N_e$ over time, the seven tree species can be divided into three major groups, highlighting the importance of life history and range size in determining synchronous variation in genetic diversity over time. Altogether, our results indicate that forest trees have been able to retain their evolutionary potential over very long periods of time despite strong environmental changes.

Extant northern temperate and boreal tree species have existed for millions of years and survived multiple glacial cycles. Micro- and macrofossil data indicate that these tree species have undergone huge range changes and large fluctuations in their census population sizes ($N_c$) over time[1]. This was largely corroborated by numerous phylogeographic studies[2]. Yet, today most tree species harbor large amounts of genetic diversity[3] and they have been shown to respond rapidly, both genetically and demographically, to recent environmental challenges such as the Little Ice Age[4] or the Last Glacial Maximum (LGM)[5,6]. While these rapid responses to new selection pressures are consistent with their current large population sizes, high realized outcrossing rates, and efficient gene flow, they may seem paradoxical in view of the large census size changes suggested by the fossil records. Unfortunately, we still lack a comprehensive view of the impact of past demographic changes on the effective population size ($N_e$), the key evolutionary parameter defining the genetic diversity and efficacy of selection[7]. For example, did $N_e$ fluctuate strongly through time or, on the contrary, was it retained and stable over repeated

A full list of affiliations appears at the end of the paper. *A list of authors and their affiliations appears at the end of the paper.

✉e-mail: pascal.milesi@scilifelab.uu.se; martin.lascoux@ebc.uu.se; tanja.pyhajarvi@helsinki.fi

glacial cycles despite changes in $N_c$? Were changes in $N_e$ primarily driven by climatic events or do they also reflect intrinsic biological characteristics such as life history or physiological properties? In the former case one would expect a high synchronicity in changes across multiple species independently of their biological properties while, in the latter case, one would expect species to form categories according to their patterns of intraspecific diversity changes and shared biological properties[8,9].

In Europe, the LGM which occurred ~27,000 to 19,000 years ago, and ensuing Holocene recolonizations have often been assumed to be the main drivers of the current distribution of intraspecific genetic diversity, with southern populations being typically more diverged than those from the more northern core range[10,11]. Earlier analyses of the demographic histories of European forest tree species generally relied on organellar markers whose polymorphisms are informative on a shorter time span than nuclear markers since in monoecious species their effective population size is half of that for nuclear markers and in dioecious a quarter[12,13]. Quite naturally, outcomes were interpreted from the perspective of only the most recent glacial period (i.e., LGM)[11]. Further, inferences based on organellar markers that behave as a single locus and, in most cases, are maternally inherited and only disperse via seeds, have limited relevance for nuclear genetic diversity wherein most of the genetic variation lies. Genome re-sequencing combined with coalescence-based demographic methods allowed inferring the demographic history of forest trees and its timescale well beyond the LGM, up to millions of years. However, most studies so far address single species or focus on inferring the timing of divergence and the extent of gene flow between populations or closely related species[14–18]. Congruence between population history and glacial oscillations has been observed in some species[14,19] but remained far from being conclusive in others[20,21]. A general conclusion on the drivers of temporal changes in genetic diversity across species cannot, however, be drawn from the compilation of these studies, or even from the reanalysis of the data they present, due to the heterogeneity of sampling strategies, genomic sources of polymorphism, and numbers of loci.

Here, we carried out a comprehensive demographic inference of seven major European tree species, distributed from the boreal to the Mediterranean regions (Table 1), based on a common strategy both for sampling populations across Europe and for sequencing genomic regions. All seven species are wind-pollinated, three are conifers (*Picea abies*, *Pinus pinaster*, and *Pinus sylvestris*) and four are angiosperms (*Betula pendula*, *Fagus sylvatica*, *Populus nigra*, and *Quercus petraea*). We conducted targeted nuclear DNA sequencing (~10,000 species-specific probes that covered ~3 Mbp of largely orthologous sequences) on a total of 3407 adult trees collected from 19 to 26 locations per species (~25 individuals each) across large parts of their natural ranges (Figs. 1, S1, Supplementary Text, Table 1, S1, Supplementary Data 1)[22]. We first conducted a comprehensive survey of the distribution of current genetic diversity in all seven species and used state-of-the-art coalescent approaches to reconstruct changes in $N_e$ over multiple glacial cycles and test for synchronous changes across species.

In this work, we show that past glacial and interglacial cycles did not have a major impact on genetic diversity of common European tree species. All seven species show signs of recent population growth and species cluster in both diversity-divergence spectrum and based on their demographic trajectories. Importantly, this last clustering does not reflect their phylogenetic closeness, but instead is likely the consequence of shared ecological and biological characteristics.

## Results
### Patterns of genetic diversity do not reflect phylogeny or environmental preferences
Overall, current genetic diversity and structure in the seven species (Fig. 1) reflected neither phylogeny nor current environmental

preferences. The patterns likely followed from a combination of biological and ecological characteristics as well as range-limit constraints (biotic or abiotic). With respect to genetic diversity and genetic differentiation among populations, the seven species can be divided into four sets: highly genetically structured and low diversity *P. pinaster* and *P. nigra*; moderately structured, and intermediate diversity *F. sylvatica* and *P. abies*; moderately structured, high diversity *Q. petraea*; and finally, panmictic and moderate diversity *P. sylvestris* and *B. pendula* (Fig. 1I).

Nucleotide diversity at four-fold degenerate, synonymous sites ($\pi_4$) ranged from 0.0027 to 0.0072 per bp across the seven species (Table 1, Supplementary data 2), as is typical of outcrossing trees[23], and was remarkably similar among populations of a given species (Fig. 1H). *Quercus petraea* had the highest genetic diversity and it increased towards the north. Also, boreal species (*P. abies*, *P. sylvestris* and *B. pendula*) exhibited slightly higher diversity at high latitudes, whereas genetic diversity tended to decrease northwards for the temperate species *P. nigra*, *F. sylvatica* and *P. pinaster* (Fig. 1H, Supplementary Data 2). Thus, the geographic distribution of genetic diversity did not consistently follow the south–north latitudinal gradient that is often considered as a proxy for postglacial recolonization history[24].

Genetic differentiation between populations ($F_{ST}$) was low for most species, except for *P. pinaster* and *P. nigra* (Table 1). Isolation-by-distance at the range scale was significant for most species, likely reflecting the distance-dependency of wind-mediated pollen dispersal over mere population-level genetic drift (Fig. 1A–G). However, the level of divergence was not uniform across the species' distributions. Generally, the most genetically divergent populations were found at southern latitudes (Fig. S2), similarly to what Petit et al.[11] found across several angiosperm trees. This result is also supported by the spatial distribution of ancestry proportions and principal component analysis (PCA) (Figs. 1A–G, S3–S23, Supplementary Data 2). Additionally, populations at higher elevations were genetically more differentiated from the rest of the range (Fig. S2, Supplementary Data 3). Globally, genetic structure coincided with the main discontinuities in the species' distributions, but with considerable variation across species (Figs. 1A–G, S3–S16).

### Main divergence events largely predate the last glacial maximum in all seven species
To compare divergence and demographic events across species, we analyzed different subsets of the data with, in each case, simple and consistent modeling choices and methodology. The origin and timing of the divergence between populations (Table 1) was studied using demographic models implemented in fastsimcoal2[25,26]. To estimate the timeframe of population separation we analyzed in each species two non-admixed populations representing the main southern and northern clusters (Supplementary Data 4, Fig. S24). In all species, divergence models with migration had better support than models without (Supplementary Data 4), and the estimated divergence times between major clusters largely predated the LGM, extending from 0.6 Mya (middle Quaternary) up to 17 Mya within the Miocene (Fig. S25, Supplementary Data 4–6). Hence, for all species the formation of the main genetic groups was an outcome of demographic events having occurred over multiple glacial cycles, and, importantly, these groups were preserved through glacial cycles despite extensive gene flow. Consequently, the overall pattern of genetic differentiation better reflects topography and other persistent barriers to movement of populations, instead of recent divergence during the LGM. For example, the mountains of southern Europe (e.g., Pyrenees, Alps) could have driven the recurrent formation of sky islands, i.e., isolated high-elevation regions to which cold-adapted species repeatedly shifted during interglacial periods, resulting in higher divergence between southern high-elevation populations than between populations at lower altitudes or populations at more northern latitudes[27].

**Table 1 | Biological characteristics and genetic summary statistics for seven European tree species**

| Species | Biological characteristics | Hybridization with | Min. flowering age (years) | Max. age known (years) | $F_{ST}$ | $\pi_4$ (per bp) | IBD (slope, $10^{-3}$) |
|---|---|---|---|---|---|---|---|
| *Betula pendula* (Silver birch) | • deciduous<br>• temperate to boreal<br>• wind pollination<br>• wind seed dispersal<br>• large and continuous range | *B. platyphylla*<br>*B. pubescens* | 10–25 | 90–150 | 0.03[a] | 0.0036 | 4 |
| *Fagus sylvatica* (European beech) | • deciduous<br>• temperate<br>• wind pollination<br>• animal seed dispersal<br>• large and continuous range | *F. orientalis* | 40–50 | 150–300 | 0.05[a] | 0.0050 | 15 |
| *Populus nigra* (Black poplar) | • deciduous<br>• Mediterranean to temperate<br>• wind pollination<br>• water and wind seed dispersal<br>• vegetative and sexual reproduction<br>• intermediate and discontinuous range (riparian) | *P. nigra* 'Italica'<br>*P. deltoides*<br>*P. trichocarpa*<br>*P. maximowiczii*<br>*Populus* sp. hybrid cultivars | 4–10 | 100–400 | 0.16[a] | 0.0032 | 125 |
| *Quercus petraea* (Sessile oak) | • deciduous<br>• temperate<br>• wind pollination<br>• animal seed dispersal<br>• large and continuous range | *Q. robur*<br>*Q. pubescens*<br>*Q. pyrenaica* | 40–100 | >1000 | 0.04[a] | 0.0072 | 9 |
| *Picea abies* (Norway spruce) | • conifer<br>• temperate to boreal<br>• wind pollination<br>• wind seed dispersal<br>• large and discontinuous range | *P. obovata* | 20–40 | 200–300 | 0.06[a] | 0.0048 | 33 |
| *Pinus pinaster* (Maritime pine) | • conifer<br>• Mediterranean<br>• wind pollination<br>• wind seed dispersal<br>• limited and discontinuous range | | 6–20 | 120–250 | 0.13[a] | 0.0027 | 59 |
| *Pinus sylvestris* (Scots pine) | • conifer<br>• temperate to boreal<br>• wind pollination<br>• wind seed dispersal<br>• large and continuous range | *P. mugo*<br>*P. uliginosa* | 15–30 | 400–750 | 0.01[a] | 0.0039 | 3 |

Age information was retrieved from the European atlas of tree species https://forest.jrc.ec.europa.eu/en/european-atlas/ and the European forest genetic resources program (EUFORGEN).
Notes: mean pairwise genetic differentiation ($F_{ST}$).
[a] one-sided *p*-value for among population variation based on AMOVA was 0.01 for all species; pairwise nucleotide diversity per site at four-fold sites ($\pi_4$); Isolation by distance (IBD) is represented by the slope of the regression of $F_{ST}$ /(1- $F_{ST}$) over the logarithm of the distance (km) between pairs of populations.

## General increase in $N_e$ over multiple glacial cycles

To infer the timescale of changes in $N_e$ in the seven species and over many glacial cycles, we used Stairway Plot 2[28], a composite likelihood method that infers changes in $N_e$ over time from site frequency spectra (SFS). Because Stairway Plot 2 is model-flexible and inferences can be biased towards complex models, with more demographic events occurring (overfitting), we tested the robustness of the results with the more constrained fastsimcoal2 2-epoch model. SFS-based inference of $N_e$ trajectory measures changes in coalescence rates of gene genealogies, which depend on historical changes in $N_c$ affecting $N_e$ but also on barriers to gene flow in structured populations and the way gene genealogies are sampled ([29] and references therein). To account for the effect of sampling and population structure on demographic inferences[30], we conducted analyses at the species, population, and one-sample-per-population levels. With the last level, the analysis was focused on the dominating collecting phase of the genealogy[31,32].

All estimates of past and present $N_e$ were in the range of tens or hundreds of thousands; these values are much smaller than any reasonable estimate of the study species' current census sizes ($N_c$), which is in the scale of billions of individuals for most species[33,34]. While the observed ratios were lower than usual estimates of $N_e/N_c$, they fit with the trend of species with large $N_c$ displaying below-average estimates of $N_e/N_c$[35].

All species, except *F. sylvatica*, showed an excess of rare variants in the SFS (Fig. 2) revealing a global signal of ancient population growth (from 0.6 Mya in *P. sylvestris* to 15 Mya in *Q. petraea*). This signal was consistent across sampling schemes and the two inference methods (Figs. 3A–B, S26) and is in line with earlier studies on, e.g., *P. abies* and *P. sylvestris*[17,36]. Strikingly, very few populations exhibited a signal of decreasing $N_e$ through time, and those populations were often disconnected from the rest of the range and, thus, likely to have experienced stronger genetic drift (Fig. S27, Supplementary Data 7). The magnitude of increase in $N_e$ varied across species and was largest for *P. sylvestris* (from ~5000 to 500,000) and weakest for *F. sylvatica*, for which the proportional increase was only two-fold (from 100,000 to 200,000).

Crucially, these patterns suggest that the overall genetic diversity of each species, or equivalently for neutral loci their $N_e$, has been maintained even during the massive range contractions caused by glacial advances. In other words, dominant forest trees with large ranges, large $N_c$, and efficient gene flow have retained genetic diversity over long periods of time, despite regional extinctions during ecologically unfavorable periods. This contrasts with previous studies that generally tended to consider the LGM as the major cause of genetic change in forest tree species in Europe. Simulations of unstructured populations undergoing cyclic 10-fold demographic changes confirm

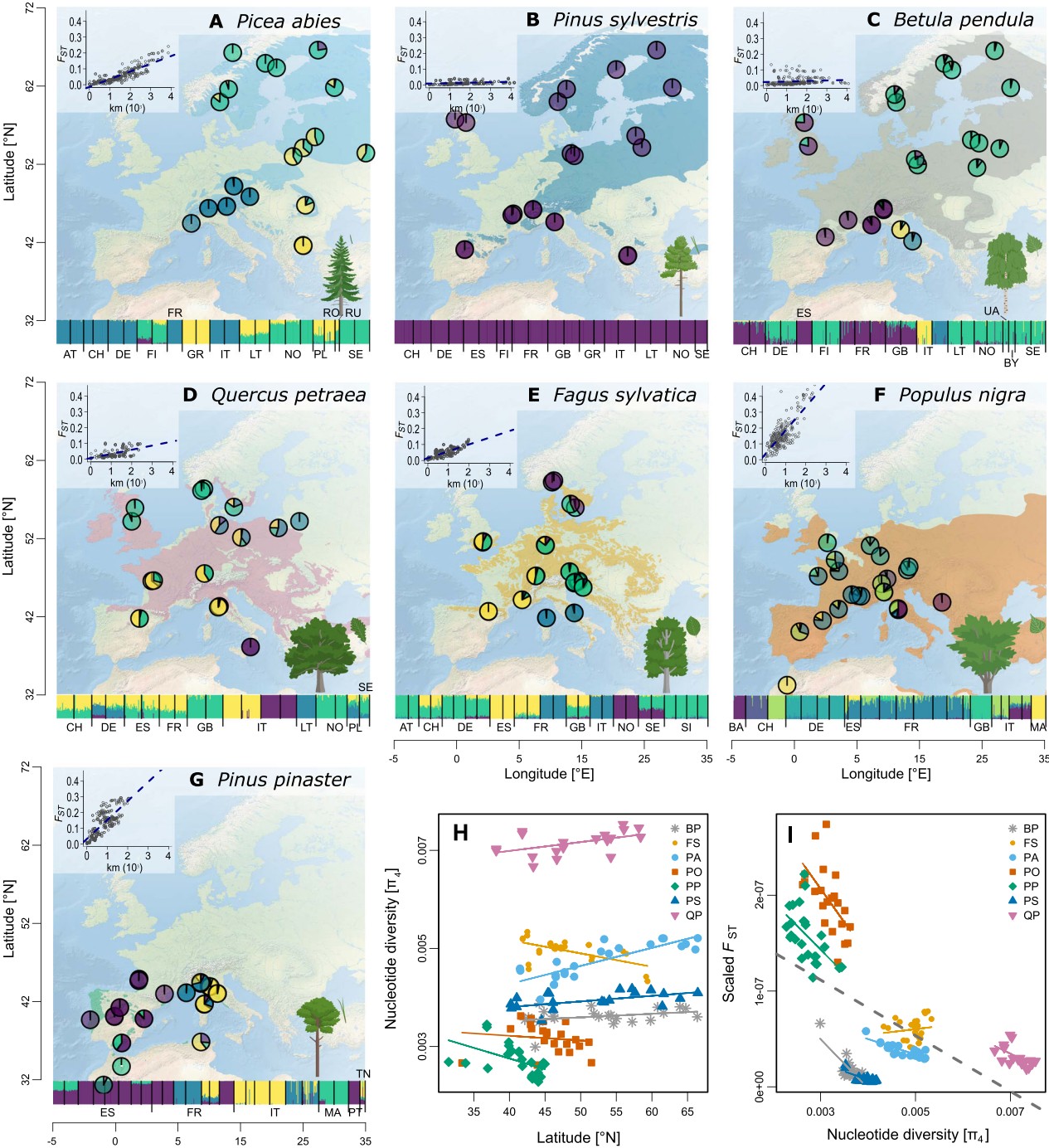

**Fig. 1 | Admixture, isolation by distance, genetic diversity and divergence patterns of the seven studied European tree species. A–G** Pie charts represent population average admixture coefficients. Four genetic clusters are shown to visualize genetic structure, except for *Pinus sylvestris* (K = 1; **B**) and *Populus nigra* (K = 7; **F**) (see Figs. S3–S16 for other cluster numbers). An admixture bar plot of all individuals is shown at the bottom of each panel (country codes are explained in Table S1). Background maps[82] represent species' ranges. Inset graphs show patterns of isolation by distance. **H** Nucleotide diversity at four-fold sites, $\pi_4$, per base pair as a function of latitude. **I** Population-specific scaled differentiation estimated as the average of the ratios of pairwise $F_{ST}$ over pairwise distance for all population pairs as

a function of $\pi_4$. Different symbols and colors represent populations of the different species with their respective trend lines. Species codes are explained in Table S1. The few populations from Russia are not represented here and can be found in Figs. S1, S12. (AT Austria, BA Bosnia and Herzegovina, BY Belarus, CH Switzerland, DE Germany, ES Spain, FI Finland, FR France, GB Great Britain, GR Greece, IT Italy, LT Lithuania, MA Morocco, NO Norway, PL Poland, PT Portugal, RO Romania, RU Russia, SE Sweden, SI Slovenia, TN Tunisia, UA Ukraine, BP *Betula pendula*, FS *Fagus sylvatica*, PA *Picea abies*, PO *Populus nigra*, PP *Pinus pinaster*, PS *Pinus sylvestris*, QP *Quercus petraea*). Source data are provided as a Source Data file-1.

that $N_c$ fluctuations do not translate in fluctuating SFS-based $N_e$ trajectories when $N_e$ remains large (>100,000) and generation time is long (>=15 years) but rather suggest an ancient decay of $N_e$ (Figs. S28–S31).

### Synchronicity and idiosyncrasy in changes in $N_e$

Across the seven species, the changes in $N_e$ over time were not entirely species-specific (Figs. 3C, S32 and Table S2). Instead, three groups of species could be distinguished based on their demographic

trajectories (Fig. 3C). A first group included the three boreal species (*P. abies*, *P. sylvestris* and *B. pendula*) and the riparian *P. nigra*, a second group comprised the two major temperate broadleaves (*F. sylvatica* and *Q. petraea*), and the only Mediterranean conifer, *P. pinaster*,

constituted a group of its own. Interestingly, this grouping differs from the one established above based on current nucleotide diversity and population genetic structure. Neither patterns of $N_e$ of individual species nor synchronous changes align well with the known glacial and interglacial periods (Fig. 3). This contrasts with some earlier results of individual forest tree species (e.g., refs. 14, 19, 37), but is in line with other comparative studies[8,9]. The overall lack of correlation between changes in $N_e$ and climatic oscillations suggest that forest tree populations remained highly interconnected and acted as a single, large metapopulation, whose $N_e$ was less affected by climatic fluctuations than $N_c$. Still, repeated synchronous phases of slight decrease in $N_e$ can be detected despite the global increasing trends, likely being the signature of shared, recurrent and possibly delayed influence of glacial cycles on these metapopulations (Fig. S32 and Table S2).

## Discussion

### Current population genetic diversity is the result of long-term processes

Our study demonstrates that these tree species have been able to retain their evolutionary potential through multiple glacial cycles. This is in agreement with recent studies showing the ability of tree species to rapidly respond to environmental challenges[4,6] and to swiftly colonize new areas as they become suitable[38]. This potential likely reflects their unique biological features. Very large and genetically connected populations along with long generation time—hence, a limited number of generations with elevated drift—allowed forest tree species to retain genetic diversity through time, despite intermittent, substantial

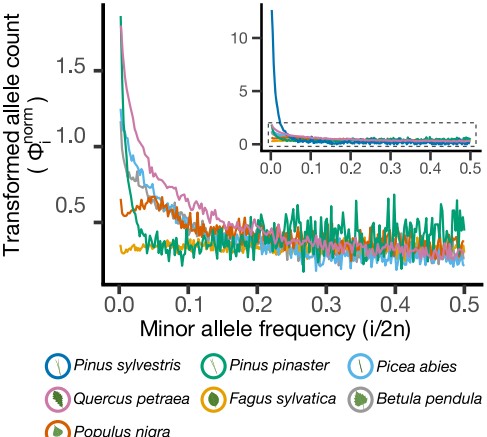

**Fig. 2 | Folded site frequency spectra (fSFS) of all seven studied species (inset) and of the remaining six species after excluding *Pinus sylvestris* (with adjusted scale in the y axis).** The SFS were transformed and normalized following[83] (see Materials & Methods) so that they are flat under the standard neutral model, facilitating the visualization of the SFS that depart from expectations of the standard neutral model[84,85]. Source data are provided as a Source Data file-1.

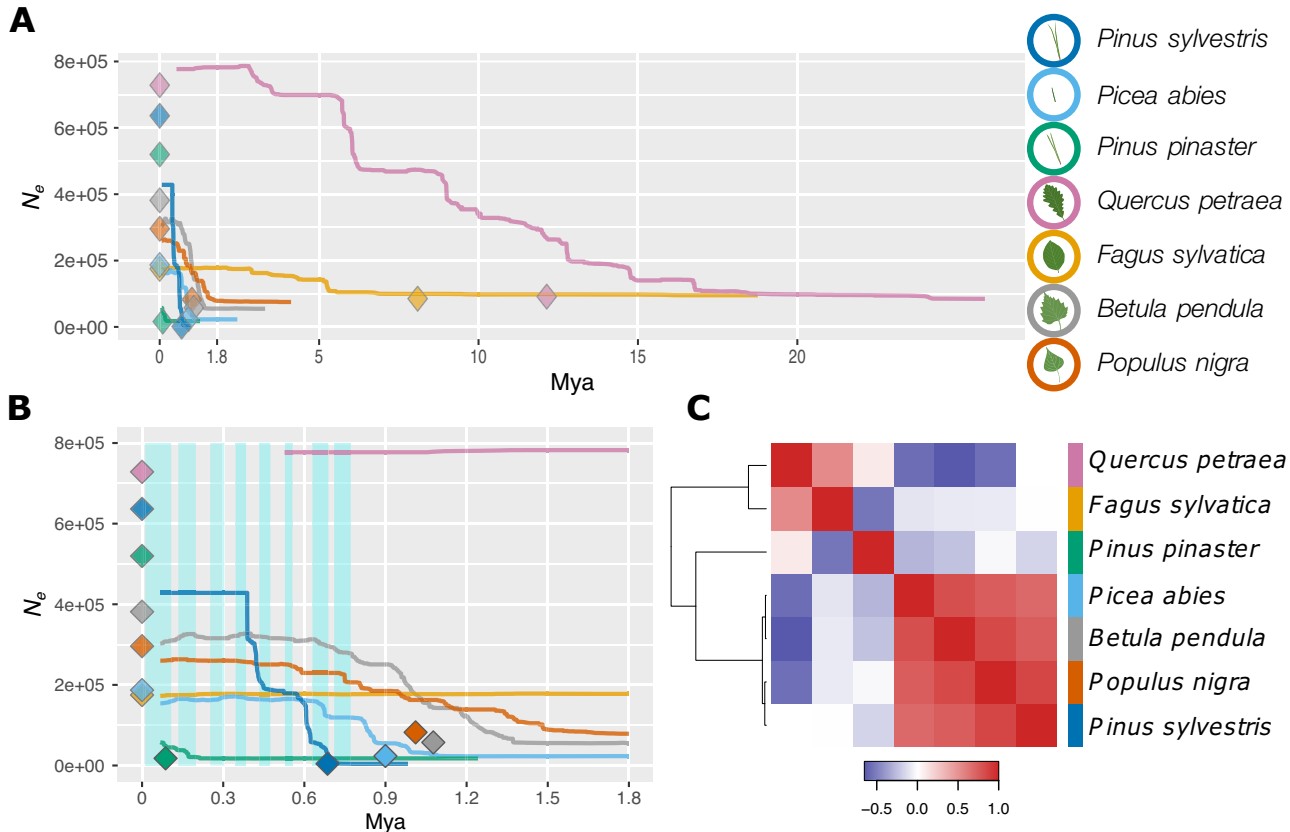

**Fig. 3 | Demographic change across time for the seven tree species. A** Change in effective population size ($N_e$) through time (million years ago, Mya), inferred with Stairway Plot 2 (lines, one-sample-per-population dataset; see Supplementary Data 7) or with fastsimcoal2 (diamonds, 2-epoch model and one-sample-per-population dataset; see Supplementary Data 5 and 6). The median changes in $N_e$ are reported from both methods. **B** A zoom-in of (**A**) focusing on the 0–1.8 Mya period. Blue

shaded rectangles delineate glacial periods. Timing of the glacial periods was inferred from composite $CO_2$ records publicly available at (https://www.ncei.noaa.gov/). **C** Heatmap based on Kendall's correlation coefficients computed from changes in $N_e$ through time between each pair of species. The order of species along the x-axis is the same as that along the y-axis. Blue and red colors represent negative and positive correlation values, respectively. Source data are provided as a Source Data file-1.

geographical range contractions. This diversity has taken shape and accumulated over very long periods of time, involving multiple glacial–interglacial cycles. Similarly, an investigation of the Distribution of Fitness Effects (DFE) of new mutations for a subset of the same data showed differences in DFE between species but no differences among populations within species. This finding also suggests that differences in DFE parameters accrue over long time periods[39] and essentially reflect the collecting phase of the coalescent process of a many-demes model[31,32]. It is worth pointing out that the seven species considered in this study are all widely distributed, relatively abundant, and ecologically important species of European forests. As such, they are interesting and important models to study the effects of environmental factors on the evolutions of European forests across space and time. However, they do not form a representative sample of the modern European tree flora. Furthermore, current species are those that survived past mass extinctions. At the Pliocene-Pleistocene transition the climate in the northern hemisphere changed dramatically with the onset of the glacial-interglacial periods, resulting in large scale extinction of trees, especially in Europe[40]. The modern flora represents less than 30% of the tree genera present during the Tertiary[41,42]. Extinctions eliminated deterministically cold sensitive species[43], and following episodes of selection further favored species sharing invasiveness attributes[44] (prolificity, competitive ability, dispersal) that facilitated locally the replacement of the extinct species.

While the use of pollen and macrofossil records as well as low-resolution uni-parentally inherited organellar DNA markers in previous studies revealed post-LGM migration patterns[11], these approaches did not give a complete understanding of the dynamics of genetic diversity over time, simply because both data types cover a too short time period. Pollen records are informative on past plant population movements but chronologies of tree pollen records are limited in time[45], often uncertain especially at the species level[46], and heterogeneous across space[47]. Further, they comprise no information on within-species genetic relationships. Long-lived organisms, such as forest trees, have fewer generations per glacial cycle than annual and herbaceous species, and therefore will, for the same number of generations, experience a larger number of glacial cycles. Simulations of demographic changes following glacial cycles show that inferred $N_e$ trajectory of annuals can capture the demographic fluctuations of the last glacial-interglacial cycle. However, it is not the case for organisms with much longer generation time like trees (Figs. S28–S31). On the evolutionary time scale, glacial cycles are shorter and recur faster for species with longer generation time and larger $N_e$. Hence, as was indeed observed, the current structure of genetic diversity reasonably reflects the impact of many glacial cycles (e.g., refs. [48], [49]). Importantly, our results explain how tree species that have survived repeated glacial cycles were able to retain genetic diversity and, hence, a capacity to respond to new environmental challenges. However, genetic stability across millions of years does not exclude drastic changes in the short term, e.g., in species distributions and local abundances, which can have major impacts on ecosystem and forest functions.

Since we focused on inferring these demographic events based on current genetic diversity, we disregarded individuals with a high degree of admixture. However, in most plant species, including trees, hybridization contributes significantly to genetic diversity[14,50,51]. Among species included here, this primarily holds for *Q. petraea*[52], which shows high nucleotide diversity (Fig. 1H) among the seven species we studied, but it may also be relevant for others that show hybridization at least in parts of their ranges or have hybridized in the past (Table 1). For demographic inference, hybridization introduces signals of even older evolutionary events and leads to elevated estimates of $N_e$. For predicting population responses to climate change, more information on groups of closely related species will be essential, especially as introgression can be important in environmental adaptation[53].

## Changes in $N_e$ are not only driven by glacial cycles

The genetic diversity of all seven species was maintained across long periods of time: most of the time, $N_e$ either increased or remained stable. However, we did not observe a single shared dynamic of changes in $N_e$ across the seven species, which would be expected if glacial cycles were the main drivers. Instead, there were three clear species groups based on $N_e$ change over time (Fig. 3C). In this respect, our results are congruent with previous studies that also showed a tendency to low levels of overall synchronicity among species and even populations within species ([8,9] and references therein). Bai et al. [8] carried out a comparative demographic study in walnut species and showed that the timing and amplitude of changes in $N_e$ differed among species. They concluded that the population histories of these walnut species were not driven by extrinsic environmental changes alone and that different species responded idiosyncratically to similar environmental challenges[8]. Similarly, ref. [9] reconstructed the trajectories of $N_e$ of three palm species and four Annonaceae tree species from African rainforest. Here too, evolutionary responses were largely asynchronous and individualistic. Interestingly, the three palm species had large $N_e$ (around 500,000) that increased regularly through time, as observed for our seven temperate forest tree species. This was in contrast to the four Annonaceae species, whose $N_e$ were significantly smaller (around 50,000) and fluctuated strongly over the same time period. In the present study, the peculiarity of *P. pinaster* likely mirrors its more southern distribution as well as its highly fragmented range compared to the other six species while the synchronicity between *Q. petraea* and *F. sylvatica* likely finds its origin in similarities in both biology (e.g., generation time, ecological niche, dispersal) and geographical distribution. The recovered demographic histories of *Q. petraea* and *F. sylvatica* go back much further in time (i.e., ~32 and ~17 Mya, respectively) than for the other studied species. The depth of the *Q. petraea* genealogy and its relatively large $N_e$ are likely a consequence of continuous hybridization with other abundant and closely related white oak species[54], combined with long generation times. *Fagus sylvatica* may have had small, secondary refugia outside the core refugial areas, maintaining local reservoirs of genetic diversity across glacial cycles[47,55]. The synchronicity of the three boreal species also reflects some similarities in both biology and geographical distribution, although they also show marked differences in population genetic estimates. One explanation for the grouping of *P. nigra* with the boreal species could be that as a riparian species it was less affected by global climatic patterns and could have survived glacial times in microrefugia close to rivers, also in colder regions. Finally, it is worth pointing out that the two pine species, *P. pinaster* and *P. sylvestris*, were very distinct from each other, despite similar mutation rates, rates of adaptive evolution and generation times, reflecting their different ecological characteristics and geographic ranges[5,56].

$N_e$ estimates and their timing scale according to the assumed mutation rates. We used the best current estimates obtained from forest trees based (see detail below). However, it is possible that the scaling of events may extend or compress across the timeline as more precise estimates of mutation rates and generation time become available. It is noteworthy that the actual $N_e$ trajectories are not affected by the mutation rate, just their scaling.

The present study highlights the existence of some commonalities in the $N_e$ trajectories of seven major European forest tree species. Firstly, the $N_e$ of all seven species we analyzed showed a mostly monotonic growth over a large part of the Quaternary. Hence, the genetic diversity of large tree metapopulations has been strikingly resilient to the drastic environmental changes during which the species experienced regional extinctions and extensive shifts in species distributions. Secondly, the trajectories of $N_e$ through time were correlated within groups of species sharing ecological and biogeographical properties, but not across all species. This supports the idea that changes in $N_e$ are not solely driven by climatic events but also reflect species'

shared biological characteristics such as life history or physiological properties. For instance, two species, such as *B. pendula* and *P. abies*, can have rather different extant population genetic structure across their similar distribution ranges, and yet have highly similar $N_e$ trajectories.

Finally, it has been suggested that understanding individual species' responses to past climatic oscillations is critical to predict their ability to cope with climate change[9]. Our results, in contrast, suggest that species' idiosyncratic responses, while highly desirable to understand for other purposes, may not be required for predicting their evolutionary response to ongoing rapid climate change. However, recovering the evolutionary response of a wider spectrum of species would be needed to establish a reliable typology of demographic histories and their effects on genetic diversity.

## Methods

### Sampling

We sampled seven tree species of considerable economic and/or ecological importance in Europe: *B. pendula*, *F. sylvatica*, *P. nigra*, *P. abies*, *P. pinaster*, *P. sylvestris* and *Q. petraea*. For each species, we sampled a minimum of 20 populations (Table S1, Supplementary Data 1) from across the species' natural distribution ranges (Fig. S1). Sampling was carried out within the framework of the EU Horizon 2020 project GenTree. The majority of the sampled populations are the same as reported in ref. 22, with additional samples reported in Table S1. Sampling principles and details are described in ref. 22. We dried the samples with silica gel and collated them in a single lab per species, where we extracted DNA (see Table S3 for details). We eluted the DNA in water, quality-checked it by UV spectrophotometry, and treated it with RNAse. We then sent all DNA extracts to IGA Technology Services (Udine, Italy) for targeted sequencing.

### Sequencing and SNP calling

We focused on a limited part of the genome (3 Mbp) using targeted sequencing. The targeted regions consisted of orthologous genes involved in putative functions of interest, species-specific candidate genes, and randomly selected genes (Table S4). We first established a list of 2639 genes involved in functions of interest (e.g., response to stress, immune response, circadian clock, detection of abiotic stimulus) in *Arabidopsis thaliana* using a term search in Gene Ontology (GO) and Kyoto Encyclopedia of Genes and Genomes (Supplementary Data 8). For each species independently, we then identified putative orthologs to those 2639 genes using a reciprocal best hit approach based on protein sequences (Blastp, BLAST v.2.5.0 +). Finally, we used Orthofinder v.1.1.4[57] on the complete set of putative orthologs to define orthogroups across the seven species. The *Quercus robur* reference genome was used to identify orthologs for *Q. petraea*, a closely related species with which it often hybridizes. The *Populus trichocarpa* reference genome was used for *P. nigra*. We selected 811 best orthogroups that included at least one gene for at least six of the seven tree species (Supplementary Data 9). For some species, we also included a variable number of other orthogroups including genes that were found only in a reduced set of species (e.g., 59 additional genes across the three conifer species; Supplementary Data 10). For each species, we then completed the ortholog list with genes of interest and randomly selected genes to reach up to 6 Mbp of sequence to serve as a template for probe definition. Starting from the 6 Mbp of sequence mentioned above, Roche designed a set of uniquely mapping probes based on either a reference genome or coding sequence (CDS) data coordinates (Table S4), relying on its custom probe design pipeline (454 Life Sciences, a Roche company, Branford, CT, USA) including the SSAHA algorithm[58]. We then restricted candidate probes to cover 3 Mbp of sequence, prioritizing probes covering best-ortholog genes (Table S5).

To estimate the quality of genomic DNA, we quantified random samples from each 96-well plate using a Qubit 2.0 Fluorometer (Invitrogen, Carlsbad, CA, USA) and a NanoDrop 1000 spectrophotometer (Thermo Fisher Scientific, Waltham, MA, USA). We quantified all 4754 samples using the GloMax Explorer System (Promega Corporation, Madison, WI, USA) and prepared libraries for target enrichment using the SeqCap EZ – HyperPlus kit (Roche Sequencing Solutions, Pleasanton, CA, USA) with 100 ng/μl of input DNA, following the manufacturer's instructions. For *P. abies* we conducted a second round of library preparation using 200 ng/μL of input DNA. We evaluated library size using the Bioanalyzer High Sensitivity DNA assay (Agilent Technologies, Santa Clara, CA, USA) and quantified libraries using a Qubit 2.0 Fluorometer. We sequenced the libraries on a HiSeq 2500 (125 cycles per read) for *P. abies* and *B. pendula* and on a NovaSeq 6000 (Illumina, San Diego, CA, USA; 150 cycles per read) for the remaining species, in both cases working in paired-end mode. We used Illumina bcl2fastq v.2.20 for base calling and demultiplexing, and we used ERNE v.1.4.6[59] and Cutadapt[60] for quality and adapter trimming, both with default parameters.

For mapping we used BWA mem v.0.7.17[61] and samtools v.1.7[62] against the available reference genome of the same or closely related species, adding the mitochondrial and chloroplast genomes when those were missing from the reference (Tables S6–S7, see https://github.com/GenTree-h2020-eu/GenTree/blob/master/Alignment_commands.txt for exact commands). In brief, we removed reads mapping on the organellar genomes, multiple mapping reads using samtools, marked duplicates with Picard and removed them. We maintained only a relevant portion of the genome with sufficient depth (>5n, where n = sample size) for the next step to speed up the SNP calling stage, which can be computationally demanding for large, fragmented genomes.

We performed SNP calling using the software package GATK v.4.0.10.0[59]. Briefly, we ran HaplotypeCaller in GVCF mode to call potential variant sites at the single-sample level, then used GenomicsDBImport and GenotypeGVCFs to perform joint genotyping on the entire cohort of samples. Command lines are available at https://github.com/GenTree-h2020-eu/GenTree/blob/master/SNP_calling_commands.txt.

### QC and SNP filtering

We conducted initial quality control of the SNP data using PCA and ADMIXTURE. We removed non-desired samples, e.g., misidentified species or obvious hybrids. We also removed samples with excessive amounts of missing data ( ≥ 60% for all species except *B. pendula* where the threshold was ≥ 20%) or extreme values of heterozygosity ( ≥ 6% of the calls heterozygous) for *P. nigra* and *B. pendula*. Jupyter notebooks of initial quality control are available at https://github.com/GenTree-h2020-eu/GenTree/tree/master/cervantesarango/JupyterNotebooks.

We retained bi-allelic variants and followed GATK recommendations to exclude poorly supported SNPs with scores QD < 0.25, QUAL < 20, SOR > 3.0, MQ < 30, MQRankSum < −12.5 and/or ReadPosRankSum < −8.0. Filtering code is available at https://github.com/GenTree-h2020-eu/GenTree/tree/master/cervantesarango/GATK_rawSNPs_to_v2. We identified putative false SNPs derived from paralogs mapping to a single location in the reference genome. We based the identification on heterozygote excess (H > 0.6) and deviation from the expected read ratio (D < −20 or D > 20) using the HDplot method https://github.com/GenTree-h2020-eu/GenTree/tree/master/rellstab[63].

We further utilized the location information of each SNP to delineate regions with an especially large proportion of paralog-derived SNPs

$$\frac{\text{number of paralog SNPs}}{\text{number of SNPs}} > 10\% \text{ within 250 bp} \qquad (1)$$

and excluded all additional polymorphic positions included in those regions (https://github.com/GenTree-h2020-eu/GenTree/tree/master/kastally/paralog_window_filtering).

In addition to the variant-level filtering described above, we applied genotype-level filtering. We reported genotype calls with depth (DP) < 8 or genotype quality (GQ) < 20 as missing data to ensure high-quality genotypes[64], and we filtered out SNPs with >50% missing calls to produce the v.5.3.2 VCF files.

To estimate the size of the portion of the genome sequenced with sufficient quality and depth across individual of each species independently (available genome, Table S7), we applied the same filtering procedure to the monomorphic positions, removing positions with DP < 8 and GQ < 20. Additionally, we excluded the same areas enriched with paralogs. The exact limit of those areas was defined at mid-distance between the last paralog and the next retained SNP.

Site-based annotation (4-fold degenerate, 2–3-fold degenerate and 0-fold degenerate sites) of detected SNPs was completed using the python script NewAnnotateRef.py available at https://github.com/fabbyrob/science/blob/master/pileup_analyzers/NewAnnotateRef.py.

SNPs were classified as intergenic, intron, stop, up and down using ANNOVAR[65] (Tables S8 and S9).

## SFS scaling

Inference of demographic history can be done using SFS[28,66]. The SFS must be estimated on a sample of a given size for all sites, however, in real datasets sample size varies among sites due to missing data. We therefore used the SNP set v.5.3.2 (see Supplementary Methods), removed SNPs with >50% of missing data in any population and down-sampled the SFS for each population or subset used for demographic inference (see demographic analyses section for details) to half the initial sample size. The source code for resampling the SFS is available at https://github.com/GenTree-h2020-eu/GenTree/tree/master/kastally/sfs_resampling. The SFSs produced were then used in downstream analyses (see below).

## Population genetic structure and isolation-by-distance

To characterize the main genetic clusters among populations, we used SNP from dataset v.6.3.2 with only putatively neutral SNPs (i.e., 4-fold degenerate sites or located in introns or intergenic regions), pruned for SNPs in high linkage disequilibrium and excluding singletons (plink v.1.9.), to compute ancestry components (Q score) for each individual using ADMIXTURE v.1.3[67]. We performed an unsupervised analysis of individual ancestry proportions based on maximum likelihood and implemented 20 replicates for each K value (1–12) to assess cross-validation errors. Then, we averaged the Q scores of individuals for each population and visualized the geographic distribution of the genetic groups identified in ADMIXTURE using the raster R package v.3.1.5[67].

We calculated genetic distances with dataset v.6.3.1 based on pairwise $F_{ST}$ values[68], implemented with the stamppFst function of the StAMPP R package v.1.6.3[69]. We then performed a hierarchical analysis of molecular variance (AMOVA), using the poppr.amova function of the poppr R package v.2.9.3[70], to assess the partitioning of genetic variation: (i) between populations, (ii) between individuals within populations, and (iii) within individuals. We tested levels of significance using the randtest function of poppr.

We quantified the pattern of isolation-by-distance (IBD) with dataset v.6.3.1 by regressing the genetic distance between pairs of populations ($F_{ST}$[68]) over the natural logarithm of the geographic distance between populations, following Rousset's[71] approach for a two-dimensional stepping-stone model.

$$\frac{\widehat{F_{ST_i}}}{\left(1 - \widehat{F_{ST_i}}\right)} = \beta \ln(x_i) + \alpha + \varepsilon_i \tag{2}$$

For each pair of populations $i$, we estimated $F_{ST}$ using vcftools (v.0.1.13)[72]. $x_i$ is the geodesic distance separating the pair of populations in km (geosphere R package v.1.5-10[73]), $\beta$ is the slope of the regression, $\alpha$ is the intercept, and $\varepsilon$ is the error term.

## Principal component analysis

In order to obtain a general picture of the main population structure of each species, we conducted a PCA with EIGENSOFT and default parameters (v.7.2.0, https://github.com/DreichLab/EIG). For each species we used all populations included in the dataset v5.3 and kept only 4-fold, intronic and intergenic SNPs pruned for high linkage disequilibrium. The whole procedure was repeated using dataset v.6.3.1 but removing the most divergent populations to visualize more subtle population structure.

## Scaling $F_{ST}$ and computing population-specific $F_{ST}$

To investigate the change in population differentiation along latitude, longitude and elevation, we computed the average population-specific $F_{ST}$ scaled by the natural logarithm of the geographic distance. For each focal population, we computed the average pairwise $F_{ST}$ divided by the average distance (km) between each pair of populations in which the focal population was involved.

## Measuring genetic diversity

We estimated the pairwise nucleotide diversity, π, for 4-fold ($\pi_4$) and 0-fold sites ($\pi_0$) as well as for silent sites ($\pi_s$, comprising intergenic, intronic and 4-fold sites). Based on the allele frequencies obtained from the SFS after down-sampling per species and per population based on SNP set v.5.3.2 (Table S10), we estimated the expected heterozygosity for each polymorphic site and summed the resulting values over all segregating sites. To obtain a π value per site, we divided this sum by the total number of sites, including monomorphic sites:

$$\pi = \frac{n}{L(n-1)} \sum_{i=1}^{S} \left(1 - \sum_{j=1}^{n} p_{ji}^2\right) \tag{3}$$

where $n$ is the sample size, i.e., the number of allele copies, $S$ is the number of segregating sites, $p_{ji}$ is the allele frequency at a polymorphic site $i$ for the $j$th allele, and $L$ is the total number of sites (polymorphic and monomorphic) in a class. Nucleotide diversity was estimated from and averaged over 1000 resampled replicates using https://github.com/GenTree-h2020-eu/GenTree/blob/master/kastally/sfs_resampling/vcf2sfs_resample.R and https://github.com/GenTree-h2020-eu/GenTree/blob/master/kbudde/pi_from_fsfs.R.

## Demographic modeling

To infer past changes in the effective population size ($N_e$), we applied two approaches based on the SFS: model-based estimates using fastsimcoal2[66] and model-free estimates with Stairway Plot 2 v.2.1.1[28]. For both approaches, we used folded and rescaled SFS (see above) based on 4-fold, intergenic and intronic sites (SNP set v5.3.2. without 0-fold sites). For the estimates using Stairway Plot 2, we applied different hierarchical levels and subsamples. First, we made stairway plots by pooling all the samples together by species, to maximize the power to detect relatively recent demographic events by using large sample sizes[74]. Considering that the SFS resulted from pooling all the samples together, different populations and especially population genetic clusters were not always equally represented, due to unequal missing data across populations and unequal representation (in terms of populations and individuals) of different clusters. To investigate the impact of hierarchical population clustering on the stairway plots, we reran Stairway Plot 2 at different hierarchical levels: (i) one sample per population (to account for unequal representation of populations) and (ii) separately for each population.

For the four broad-leaved species, we used the mutation rate of $7.77 \times 10^{-9}$ per site per generation[75] estimated for *Prunus*, which is close to estimates of the mutation rate per generation of $7 \times 10^{-9}$ for *A.*

*thaliana*[76] and *Silene latifolia*[77]. To scale estimates of timing from generations to years, we assumed a generation time of 60 years for *Q. petraea* and *F. sylvatica*, and 15 years for *P. nigra* and *B. pendula*. For the three conifers, we used a mutation rate of $2.7 \times 10^{-8}$ per site per generation[78] which is consistent with earlier, divergence-based estimates, assuming a 25-year generation time[17,79]. We ran all stairway plot estimates using 67% of the sites for training and 200 resamplings from the SFS. We calculated breakpoints following the suggestions in the stairway plot manual, i.e., at n/4, n/2, n*¾ and n-2 with n indicating the sample size.

To confirm the results obtained with Stairway Plot 2, we used fastsimcoal2[25,66] to infer the past demographic history of each species. We used the same folded SFSs (fSFS) as for Stairway Plot 2. We explored three single population models (Fig. S24): an equilibrium model (standard neutral model, SNM), a model with one demographic change (epoch-2) and a model with two demographic changes (epoch-3). For all parameters ($N_e$ at each step and each time of events, specifically: NCUR = most recent $N_e$; NANC = ancestral $N_e$; NBOT = $N_e$ after the first demographic event in the 3-epoch model; TBOT = time of the first demographic event; TENDBOT: time of the second demographic event in the 3-epoch model), we set a log prior with a range of 10 to $10^7$. We first inferred the best model by running 100 independent runs for each of the three models using fastsimcoal2 with $10^6$ simulations (-n 1,000,000), a minimum of 10 conditional maximization algorithm (ECM) cycles for likelihood computations (-l 10), 40 ECM cycles for parameter estimations (-L 40), and a minimum of one allele count for parameter estimation (−minSFScount 1; default value). For each model, we then identified the run with the best likelihood score. We computed the AIC score for each run, and we compared the AIC values of different models. Finally, following Excoffier et al.[66], we computed confidence intervals for each parameter, selecting the parameter values of the best run for both 2-epoch and 3-epoch models to simulate 100 SFSs using fastsimcoal2. We then ran, for each of those 100 SFSs, 100 independent runs of fastsimcoal2 using the same settings as before, obtaining 100 sets of parameters for each model, which we used to compute confidence intervals for each model parameter. We used this same approach for each species, using the SFS computed over all samples and the SFS computed using only one haplotype per population.

To further test if more realistic models would better fit the observed data and impact our inference of past demographic changes, we tested four models of demographic changes including events of divergence. For these models, we subset, for each species, samples from two locations, a southern and a northern one (we used admixture results to make sure that the samples came from two distinct gene pools; Figs. S3−S16). From these sets, we explored the same three models, with no divergence events and pooling all samples together regardless of origin, and four models of divergence differing regarding whether migration was possible after the divergence and including or not a demographic change before the divergence event. We used fastsimcoal2 to assess models, using the fSFS of all samples pooled together for reference models, and the joint fSFS of the population pairs for models of divergence. We conducted the simulations and inference with fastsimcoal2 by running 100 independent runs with 100,000 simulations each (-n 100,000), the same number of ECM cycles as used previously (-l 10 -L 40) and in folded mode (-m). We explored all parameters (including all or some of the following for the divergence models: NCUR = $N_e$ before the split but after the demographic event; NANC = ancestral $N_e$; NPOP1 = $N_e$ after the split for population 1; NPOP2 = $N_e$ after the split for population 2; TDIV = time of the split; TSEP = time of the demographic event; N1M21 = effective migration rate from population 1 to population 2; N2M12 = effective migration rate from population 2 to population 1; and the same parameters as above for the panmictic models) with a logunif prior between 10 and $10^9$, except for the parameters related to the number of effective migrants (the product between the migration rate and the effective population size, $N_e \times m$), which used logunif priors set between $10^{-4}$ and $10^4$. Finally, we compared the seven models using the likelihood and AIC score of the single best run out of 100, to identify the best model for each species and its associated parameters.

To test whether our results may be explained by a lack of statistical power, we tested the ability of Stairway Plot 2 to recover demographic cycles for a set of parameter values relevant to our data and model species: generation time (1, 15, 25 and 60 years per generation), size of the sampled genome (1 Mbp or 6 Mbp), mutation rates ($2.7 \times 10^{-8}$ or $7.7 \times 10^{-9}$ per site per generation), and current $N_e$ ($10^3$, $10^5$, $10^6$; the latest demographic event being an expansion, it is the highest $N_e$). The intensity of the expansion/contraction of population size was set to 10-fold, and we implemented 10 events (i.e., 5 cycles of an expansion event followed by a contraction event) following the expected times of the glacial / interglacial periods (i.e., a demographic expansion 15 kya, a decline 120 kya, then following a period of 120 kya, expansion or decline events successively, until a final decline about 1 Mya). Each event was modeled as an instantaneous change of $N_e$ with $N_e$ remaining stable in-between. These models were simulated with fastsimcoal2 to produce fSFS using a sample size of 20 haploid genomes. We then ran demographic analyses with Stairway Plot 2 on the simulated fSFS using the settings used with the empirical fSFS (4 breakpoints fixed at sample sizes following manual instructions and 200 simulations) (Figs. S28−S31). We also report nucleotide diversity (π) of each fSFS generated.

To assess the influence that unaccounted for population genetic structure may have on our analyses, we compare a set of three demographic inferences with Stairway plot 2 for each of the seven species. For the first one we artificially mixed two populations, the second and third ones correspond to the analysis of each population separately (Fig. S33). For each species, we selected the same pairs of populations as for the demographic modeling analyses (see above). In most cases, the inference is consistent across the mixed and separate analyses, suggesting that the level of population genetic structure we have in our data has limited effects.

## Synchronicity analysis

We then assessed whether the dynamics of changes in $N_e$ over time were species-specific or synchronous across species. To do so, we first computed Kendall's correlation coefficients between the output of Stairway Plot 2 for each pair of species and then investigated their covariance across the seven species using a heatmap (heatmap3 R package, v.1.1.9 https://cran.r-project.org/package=heatmap3 with default parameters, Fig. 3C). This global approach allowed us to quantify the synchronous pattern in change in $N_e$ over time between some species and test whether it was primarily driven by the global increase in $N_e$ over the period studied. This approach captures the main trend but will not allow the identification of specific periods of high synchronicity between several species. In particular, it will miss decreases in $N_e$ that could be expected to be associated with glacial periods. To specifically test whether periods over which the species experienced a decrease in $N_e$ showed higher synchronicity than expected given the actual change in $N_e$ over time for the various species, we used a randomization approach. This analysis was conducted independently for the two groups of species showing the highest synchronicity *F. sylvatica* and *Q. petraea* on the one hand, and *P. abies*, *B. pendula*, *P. sylvestris* and *P. nigra*, on the other hand.

More specifically, the Stairway Plot 2 output consists of a succession of intervals, defined by a given $N_e$ estimate and two different time points. To simplify this, for each species independently, each interval was represented by a pair of values: the unique $N_e$ value characteristic of the interval and the midpoint of the two time points. We excluded the most recent time point of the output of Stairway Plot 2 as it does not correspond to a proper step. As the time points at which Stairway Plot 2

provides estimates of $N_e$ are species-specific, we inferred $N_e$ at every time point in the joint dataset by considering for each species independently the value of the closest $N_e$ estimated by Stairway Plot 2. To smooth random fluctuations between time points and to mitigate the effects associated with small deviations to our estimates of generation time and mutation rate, we averaged the changes in $N_e$ between two time points ($\Delta N_e = N_{e(t)} - N_{e(t+1)}$) over 250 consecutive time points, using sliding windows ($\mu \Delta N_e$). Finally, we used a randomization approach to detect periods of time during which the pattern of synchronicity in $N_e$ change is stronger than what would be expected given species-specific change in $N_e$ over time; i.e., periods over which a larger number of species experienced a decrease in $N_e$ than expected if changes in $N_e$ were independently distributed across time in each species. For each species independently, we first randomized the vector of $\Delta N_e$ values and averaged the randomized values over 250 consecutive time points using sliding windows ($\mu \Delta N_e$), as we did for the observed data. From the randomized time series, we recorded the maximum number of species experiencing a decrease simultaneously, as well as the longest span over which synchronicity was conserved (i.e., the maximum number of consecutive positive $\Delta N_e$). We repeated the whole procedure 10,000 times and compared the observed values with the 95% percentile of the distribution of the maximum values obtained through the 10,000 simulations (Table S2 and Fig. S32).

### Reporting summary

Further information on research design is available in the Nature Portfolio Reporting Summary linked to this article.

## Data availability

The short read data generated in this study have been deposited to NCBI BioProjects under accession codes PRJNA602465, PRJNA602466, PRJNA602467, PRJNA602468, PRJNA602470, PRJNA602471, PRJNA602473. The vcf, ped and map -files generated in this study are available in Data INRAE at https://doi.org/10.57745/DV2X0M[80]. Source data are provided as a Source Data file. Source data are provided with this paper.

## Code availability

Code is available at: https://doi.org/10.5281/zenodo.7943876[81].

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

## Acknowledgements

Computations were made possible using resources from projects SNIC 2017/7-328, SNIC 2020/15-107 and SNIC 2021/5-540 of the Swedish National Infrastructure for Computing (SNIC) at UPPMAX, partially funded by the Swedish Research Council through grant agreement no. 2018-05973. The CSC – IT Center for Science (Finland), the Genotoul Bioinformatics platform at Toulouse (France) and the Genetic Diversity Centre at ETH Zurich (Switzerland) are acknowledged for generous computational and storage resources. This work also benefited from the Scientific Compute Cluster at GWDG, the joint data center of the Max Planck Society for the Advancement of Science (MPG) and the University of Göttingen (Germany). We also thank all GenTree partners, the Slovenian Forestry Institute (Slovenian Research and Innovation Agency grant P4-0107 and LIFEGENMON) and the Natural Resources Institute Finland (LUKE) for their contribution to sampling. This work was funded by European Union's Horizon 2020 Research and Innovation Programme grant agreement no. 676876; Academy of Finland grant nr. 287431 (T.P.); Swiss Secretariat for Education, Research and Innovation (SERI) contract no. 6.0032 (F.G.); Spanish Ministry of Agriculture, Fisheries and Food (MAPAMA) grant no. AEG 17-048 (D.G.); European Agricultural Fund for Rural Development (EAFRD) (D.G.); Swedish Research Council for Sustainable Development (FORMAS) grant nos. 2016-00780, 2020-01456 (M.L.); Spanish Ministry of Economy and Competitiveness (MINECO) contract n. PTA2015-10836-I (S.O.); LIFEGENMON (LIFE13 ENV/SI/000148) (M.W.); The Slovenian Research Agency, research core funding no. P4-0107 (M.W.); Biocenter Oulu (S.C.).

## Author contributions

Conceptualization: St.C., S.C.G.M., D.G., M.L., P.M., T.P., and G.G.V. Methodology: K.B.B., P.F.R., S.C.G.M., C.K., M.L., P.M., S.P., T.P., C.R., and S.S. Software: K.B.B., B.D., C.K., S.P., T.P., S.S., and I.S. Validation: Sa.C., V.J., P.M., T.P., and S.S. Formal Analysis: F.B., K.B.B., Sa.C., B.D., P.F.R., S.C.G.M., D.I.O., V.J., C.K., I.L.K., P.M., L.O., S.P., C.P., T.P., C.R., O.R., and I.S. Investigation: F.B., S.C.G.M., D.G., F.G., V.J., M.L., I.L.K., P.M., S.O., L.O., C.P., T.P., C.R., I.S., and M.W. Resources: B.D., P.F.R., S.C.G.M., D.G., F.G., M.L., S.O., L.O., C.P., C.R., and I.S., Data curation: B.D., Sa.C., V.J., C.K., P.M., L.O., and T.P. Visualization: B.D., C.K., P.M., I.S., and M.W. Funding acquisition: St.C., B.F., S.C.G.M., D.G., F.G., M.L., S.O., L.O., T.P., G.G.V., and M.W. Project administration: St.C., B.F., S.C.G.M., F.G., M.L., L.O., T.P., and G.G.V. Supervision: F.B., S.C.G.M., F.G., V.J., M.L., L.O., C.P., T.P., C.R., G.G.V., and M.W. Writing – original draft: K.B.B., Sa.C., B.D., S.C.G.M., V.J., C.K., M.L., P.M., L.O., S.P., T.P., C.R., O.R., and I.S. Writing – review & editing: F.B., K.B.B., B.D., B.F., P.F.R., S.C.G.M., F.G., V.J., C.K., M.L., I.L.K., P.M., L.O., C.R., S.S., I.S., G.G.V., and M.W.

## Funding

## Competing interests

The authors declare no competing interests.

## Additional information

¹Department of Ecology and Genetics, Evolutionary Biology Centre, Uppsala University, Uppsala, Sweden. ²SciLifeLab, Uppsala University, Uppsala, Sweden. ³Department of Forest Sciences, University of Helsinki, Helsinki, Finland. ⁴Viikki Plant Science Centre, University of Helsinki, Helsinki, Finland. ⁵Biodiversity and Conservation Biology, Swiss Federal Research Institute WSL, Birmensdorf, Switzerland. ⁶Department of Ecology and Genetics, University of Oulu, Oulu, Finland. ⁷Biocenter Oulu, University of Oulu, Oulu, Finland. ⁸Institute of Biosciences and Bioresources, National Research Council of Italy (IBBR-CNR), Sesto Fiorentino, Italy. ⁹Department of Forest Genetics and Forest Tree Breeding, Georg-August-University Goettingen, Göttingen, Germany. ¹⁰Department of Forest Genetic Resources, Northwest German Forest Research Institute, Hann. Münden, Germany. ¹¹UK Centre for Ecology & Hydrology (UKCEH), Bush

Estate, UK. ¹²INRAE, URFM, Ecology of Mediterranean Forests, Avignon, France. ¹³University of Paris-Saclay, INRAE, EPGV, Evry, France. ¹⁴University of Bordeaux, INRAE, BIOGECO, Cestas, France. ¹⁵Institute of Forest Sciences (ICIFOR), INIA-CSIC, Madrid, Spain. ¹⁶INRAE, ONF, BioForA, Orléans, France. ¹⁷Helix Venture, Mérignac, France. ¹⁸Department of Forest Biodiversity, Norwegian Institute of Bioeconomy Research (NIBIO), Aas, Norway. ¹⁹Plant Ecology and Geobotany, Philipps-Universität Marburg, Marburg, Germany. ²⁰Institute of Applied Genomics (IGA), Udine, Italy. ²¹IGA Technology Services S.r.l., Udine, Italy. ²²Slovenian Forestry Institute, Ljubljana, Slovenia. ²³These authors contributed equally: Pascal Milesi, Chedly Kastally, Benjamin Dauphin, Sandra Cervantes, Martin Lascoux, Tanja Pyhäjärvi. ✉e-mail: pascal.milesi@scilifelab.uu.se; martin.lascoux@ebc.uu.se; tanja.pyhajarvi@helsinki.fi

## the GenTree Consortium

Pascal Milesi [1,2,23] ✉, Chedly Kastally [3,4,23], Benjamin Dauphin [5,23], Sandra Cervantes [6,7,23], Francesca Bagnoli [8], Katharina B. Budde[9,10], Stephen Cavers [11], Bruno Fady[12], Patricia Faivre-Rampant [13], Santiago C. González-Martínez [14], Delphine Grivet [15], Felix Gugerli [5], Véronique Jorge [16], Isabelle Lesur Kupin[14,17], Dario I. Ojeda[18], Sanna Olsson [15], Lars Opgenoorth [5,19], Sara Pinosio [8,20], Christophe Plomion[14], Christian Rellstab [5], Odile Rogier [16], Simone Scalabrin [21], Ivan Scotti[12], Giovanni G. Vendramin [8], Marjana Westergren [22], Martin Lascoux [1,2,23] ✉ & Tanja Pyhäjärvi [3,4,23] ✉

