## [Peer Review File · Nature Communications]

REVIEWER COMMENTS

Reviewer #1 (Remarks to the Author):

This paper presents a fascinating window into the historical demographics and genetics of seven European tree species, including both conifers and angiosperms. Using targeted sequence capture of a subset of BUSCO genes, and focussing on 4-fold putatively neutral sites, using Stairway2 the authors estimate relatively robust effective population sizes over millions of years, and little to no signal from Pleistocene glaciations. This analysis is strong based on the inclusion of many species and populations, all sampled and analyzed consistently, resulting in comparable estimates of past and current demographics. It is encouraging that these species have maintained relatively large population sizes through many major environmental changes and range shifts. As such, I think it will be of widespread interest and is suitable for publication in Nature Communications after some revisions.

I previously reviewed this manuscript for another journal, and am satisfied that most of the issues I raised then have been addressed. The longer format of the current paper allows for a more thorough discussion of results, and some of the earlier inferences from the results have either altered or removed. As such, most of my points below are quite minor.

However, there is one issue I didn't previously recognize that I think needs some better justification. The time of past changes in N_e is inferred using a mutation rate to estimate number of generations, and then an assumed generation length for each species. The mutation rate for the angiosperms is based on an estimate from seed parent to offspring in *Prunus*, and is 7×10^{-9} . The rate for conifers is based on an estimate of somatic mutation rate from *Picea sitchensis* (lines 531-536). The latter is based on somatic mutation in old-growth trees that are exceedingly tall and hundreds of years old, whereas generation length for the conifers is assumed to be 25 years. It seems this high mutation rate may result in the number of generations and number of years being underestimated. I wonder if this has resulted in the compression of the timeframe of N_e trajectories in Figure 3A and 3B. I would like to see a better justification of this choice in the methods, or discussion in the paper of possible biases resulting from this choice of mutation rates, and the different bases for the estimates in angiosperms (mutations during reproduction) and conifers (somatic mutations).

Minor comments are below:

l. 48 “contrasting”, not “contrasted”

l. 51 delete “Hence,”

l. 57 – “have been” instead of “were”

l. 68 – “respond rapidly” – genetically or demographically?

l. 73 – define N_e , effective population size. First use.

l. 84 – “northern core-range” – should this be “more northern”?

l. 91 – “where” – change to “wherein”

Figure 1 – the pale, pastel colours of some species distributions don't show up well in A, E and G. Consider using the same colour for each species' range in the separate figures.

l. 149 – “similar to what Petit et al. found” – Can say “similar to Petit et al.” and specify for what species or what number of species.

- l. 168 – change “reflects” to “better reflects”
- l. 188 – “common mean estimates” – what does common mean here?
- Figure 2 – it is fascinating how different the folded site frequency spectra are for the seven species. No change needed – just a comment. What is the yellow dot to the left of the species legend?
- l. 243 – change “tree species” to “these tree species”.
- l. 272 – change “This situation” to “Amongst species included here, this” or something like that to make it clear you are referring to the current study.
- l. 275 – “of their ranges, or have hybridized in the past”
- l. 284 – change to “three clear species groups”
- l. 316 – change “commonalities” to “some commonalities”
- l. 339 – spell out names of genera.
- l. 408-409 – There must be a typo here – “extreme values of heterozygosity ($\leq 6\%$ of the calls heterozygous). Should be 60%?”
- l. 439-431 “To estimate the size of the genome with sufficient quality and depth for each species” – do you mean to estimate the size of the portion of the genome sequenced? Or the proportion of the genome sequenced in each species?
- l. 453-454 – “To characterize the main genetic clusters among populations, we used SNP from dataset v.6.3.2 with only putatively neutral SNPs (i.e., 0-fold degenerate sites . . .)” = should be 4-fold consistent with l. 487-488?

Reviewer #2 (Remarks to the Author):

The authors compared the demographic history of seven widespread European tree species to infer the impact of multiple glacial cycles on their current genetic diversity using genomic data. To this end, they used site frequency spectra (SFS) to infer historical changes in effective population size (N_e) and tested alternative demographic scenarios using coalescent simulations. Despite the strong demographic impacts climate changes must have had on the studied species, there was no strong genomic signatures on N_e , which seems to have remained high, or has increased, during the Quaternary. Inferred N_e changes show time correlation between a few species but not over all of them, suggesting that species have responded in different ways to past environmental changes. In addition, evidence of past population fragmentation are typically much older than the last glacial maximum (LGM), showing that the genetic structure we see today cannot be interpreted considering solely the fragmentation-recolonization events that occurred over the last glacial cycle. The authors discuss the possible reasons that could explain the absence of a strong signal on N_e despite evidence of strong demographic changes in terms of census population sizes.

The originality of the study is that it compares seven tree species using a large amount of genomic data (sampling a few hundreds of individuals from 19 to 26 populations per species). In a sense, this study explores again the topic addressed in a famous article of Petit et al. (2003, cited by the authors) who compared the structure of 23 European species with chloroplast markers, using here many nuclear genes, which should provide more robust estimates (plastid genomes represent a single realization of stochastic evolutionary processes). Data analyses follow state-of-the-art procedures and I mostly agree with the conclusions of the authors. However, I have some major comments.

1) The work presented does not provide evidence that the data used are powerful enough to detect the cyclic demographic changes expected from glacial cycles. Therefore, I suggest to conduct coalescent simulations of a population going through repeated high and low N_e (e.g. 10 such cycles), adjusting the number of generations spent under high or low N_e to the typical duration of glacial and interglacial periods and to the range of generation times estimated for the seven tree species, considering different combinations of high and low N_e . Simulated SFS data using the ranges of sample sizes and numbers of SNP's of the studied species could then be analysed using Stairway Plot 2 to confirm that, in principle, data are informative enough to detect cyclical changes of N_e . I wonder if a Stairway plot would detect demographic oscillations whenever N_e remains fairly high (say >1000 or 10000) during the low N_e phases, even if the N_e during high N_e phases is several orders of magnitude larger.

2) I'm concerned with the interpretation of SFS data in terms of past N_e . Fundamentally, the method used to infer past N_e from SFS consists in estimating how rates of coalescence vary through time, and this can indeed be interpreted in terms of N_e (basically the inverse of coalescence rate) assuming an isolated and panmictic population. The problem is that the studied species do not form a single isolated panmictic population but are structured in multiple populations. Most of the recent literature on demographic inference based on SFS makes the approximation that population structure can be ignored and directly convert estimated coalescence rates into N_e but this is problematic, as shown for example by Mazet et al. (2016), Chickhi et al. (2018) or Arrondondo et al. (2021). The current literature reminds me the situation three decades ago when many articles published estimates of the number of migrants per population (N_m) based on F_{st} , until a famous article by Whitlock & McCauley (1999) recalled that the conditions under which $F_{st} = 1/(1+4N_m)$ are very restrictive (island model at drift-migration equilibrium and low mutation rate) and probably not met in most real conditions. I think the scientific literature today is a bit in the same situation with SFS and past N_e estimates. Therefore, I believe the present ms could do a better job by controlling the impact of population structure. To this end, first I suggest to interpret the Stairway plots in terms of IICR (inverse instantaneous coalescence rate) rather than in terms of past N_e , as recommended by Chickhi et al. (2018). Second, additional coalescent simulations of a structured population would help assess whether population structure itself is not contributing to a certain stability of the IICR across glacial cycles. This could be done by simulating over multiple glacial cycles a system of 8 populations: 4 permanent populations corresponding to refuge areas (which might be larger during glacials than interglacials) would each be connected to one of 4 unstable populations that are very large during interglacials and small (or vanish) during glacials. Gene flow would occur in a stepping-stone manner only between the unstable populations during interglacials, and between each of the latter populations with its source (refugial) population. The four isolated refugial populations might generate the population structure observed in the majority of studied species (four genetic clusters). The resulting SFS and IICR estimates from Stairway plots could then be compared with the one obtained for a single population undergoing similar overall demographic changes. I believe such simulations could help interpret the data and may help explain the stability of stability of the IICR across glacial cycles.

3) Table 1 and p. 13: the reciprocal of the slope of pairwise $F_{st}/(1-F_{st})$ against $\log(\text{distance})$ is interpreted as a neighbourhood size ($N_s = 4\pi D\sigma^2$). This is correct at drift-dispersal equilibrium but given the large spatial scale of the sampling, the detection of different genetic clusters and the history of populations subject to glacial cycles, drift-dispersal equilibrium is probably not reached at the studied scale. Nevertheless, the slope of pairwise $F_{st}/(1-F_{st})$

against $\log(\text{distance})$ remains a useful synthetic measure of the intensity of the isolation-by-distance pattern. Hence, I suggest reporting these slopes instead of biased estimates of N_s .

4) Line 492. “To investigate the change in population differentiation along latitude, longitude and elevation, we computed the average population-specific F_{ST} scaled by geographic distance. For each focal population, we computed the average pairwise F_{ST} divided by the average distance (km) between each pair of populations in which the focal population was involved.” The proposed scaling would be justified if pairwise F_{ST} ’s tended to increase linearly with the spatial distance. However, as shown in Fig 1, pairwise F_{ST} ’s tend to increase more or less linearly with the $\log(\text{spatial distance})$. Hence, I think F_{ST} scaling should use $\log(\text{distance})$ instead of distance.

Minor comments:

Line 247: “Very large and genetically connected populations allowed tree species to retain genetic diversity through time, despite intermittent, substantial geographical range contractions.” The long generation time of trees, implying a limited number of generations of higher drift when populations are small, is probably an important feature limiting the loss of genetic diversity.

Fig. S26: Inset figures are too small to interpret. I suggest to present them in 2 or 3 pages.

Table S9: duration seems to be expressed in centuries rather than in Kya.

Table S17: “the portion of the reference genome in which at least 50% of the samples of the given species have a minimum coverage of 8x and a minimum genotype quality value of 20.” Does this refers to the last column (“Available genome”) and is it expressed in bp?

Cited references:

Arredondo, A., Mourato, B., Nguyen, K., Boitard, S., Rodríguez, W., Noûs, C., Mazet, O., & Chikhi, L. (2021). Inferring number of populations and changes in connectivity under the n-island model. *Heredity*, 126(6), 896–912. <https://doi.org/10.1038/s41437-021-00426-9>

Chikhi, L., Rodríguez, W., Grusea, S., Santos, P., Boitard, S., & Mazet, O. (2018). The IICR (inverse instantaneous coalescence rate) as a summary of genomic diversity: Insights into demographic inference and model choice. *Heredity*, 120(1), 13–24. <https://doi.org/10.1038/s41437-017-0005-6>

Mazet, O., Rodríguez, W., Grusea, S., Boitard, S., & Chikhi, L. (2016). On the importance of being structured: Instantaneous coalescence rates and human evolution--lessons for ancestral population size inference? *Heredity*, 116(4), 362–371. <https://doi.org/10.1038/hdy.2015.104>

RESPONSE TO REVIEWERS' COMMENTS

Reviewer #1 (Remarks to the Author):

This paper presents a fascinating window into the historical demographics and genetics of seven European tree species, including both conifers and angiosperms. Using targeted sequence capture of a subset of BUSCO genes, and focussing on 4-fold putatively neutral sites, using Stairway2 the authors estimate relatively robust effective population sizes over millions of years, and little to no signal from Pleistocene glaciations. This analysis is strong based on the inclusion of many species and populations, all sampled and analyzed consistently, resulting in comparable estimates of past and current demographics. It is encouraging that these species have maintained relatively large population sizes through many major environmental changes and range shifts. As such, I think it will be of widespread interest and is suitable for publication in Nature Communications after some revisions.

I previously reviewed this manuscript for another journal, and am satisfied that most of the issues I raised then have been addressed. The longer format of the current paper allows for a more thorough discussion of results, and some of the earlier inferences from the results have either altered or removed. As such, most of my points below are quite minor.

Answer: Thank you for your positive comments and previous suggestions for improvement. We agree that the manuscript is now clearer and more thorough, and we further revised and complemented the now evaluated version according to both reviewers' comments.

However, there is one issue I didn't previously recognize that I think needs some better justification. The time of past changes in N_e is inferred using a mutation rate to estimate the number of generations, and then an assumed generation length for each species. The mutation rate for the angiosperms is based on an estimate from seed parent to offspring in *Prunus*, and is 7×10^{-9} . The rate for conifers is based on an estimate of somatic mutation rate from *Picea sitchensis* (lines 531-536). The latter is based on somatic mutation in old-growth trees that are exceedingly tall and hundreds of years old, whereas generation length for the conifers is assumed to be 25 years. It seems this high mutation rate may result in the number of generations and number of years being underestimated. I wonder if this has resulted in the compression of the timeframe of N_e trajectories in Figure 3A and 3B. I would like to see a better justification of this choice in the methods, or discussion in the paper of possible biases resulting from this choice of mutation rates, and the different bases for the estimates in angiosperms (mutations during reproduction) and conifers (somatic mutations).

Answer: Indeed, for scaling the demographic inference, we need to provide the mutation rate per generation and generation time even though they are both hard to estimate accurately for long-lived trees.

The published mutation rate estimates of conifers, based on divergence and fossil calibration, range from \$0.7\$ to \$1.3 \times 10^{-9}\$ per site per year. The estimate obtained from somatic mutations of *Picea sitchensis* is \$2.7 \times 10^{-8}\$ per site per generation and falls well in this scale when divided by the generation time of 25 years, but is much smaller if a longer generation time is used, as the reviewer and authors (Hanlon et al. 2019) point out. We agree that the estimate based on somatic mutations of *P. sitchensis* alone is not an optimal choice and we have now reformulated that part, and in addition to Hanlon et al. (2019) we refer to Chen

et al. (2019) and Willyard et al. (2007). The estimate, e.g., in Chen et al. 2019 (2.75×10^{-8}), is very close to the one we used here. Thus, the mutation rate used for conifers is not based only on the somatic mutation rate, but combined information from both somatic mutation rate and divergence-based mutation rate estimates.

We agree with the reviewer that mutation rate estimates still have some uncertainty and can have an impact on the timing of events. We have modified methods and discussion text accordingly and now clarify that the mutation rate was based on combined information from several studies.

Methods:

“For the three conifers, we used a mutation rate of 2.7×10^{-8} per site per generation⁷⁵ which is consistent with earlier, divergence-based estimates, assuming a 25-year generation time^{17,76}.”

Discussion:

“ N_e estimates and their timing scale according to the assumed mutation rates. We used the best current estimates obtained from forest trees based (see detail below). However, it is possible that the scaling of events may extend or compress across the timeline as more precise estimates of mutation rates and generation time become available. It is noteworthy that the actual N_e trajectories are not affected by the mutation rate, just their scaling.”

Minor comments are below:

I. 48 “contrasting”, not “contrasted”

Done.

I. 51 delete “Hence,”

Done.

57 – “have been” instead of “were”

Done.

I. 68 – “respond rapidly” – genetically or demographically?

Answer: Actually both, it is now written explicitly in the main text.

I. 73 – define N_e , effective population size. First use.

Done.

I. 84 – “northern core-range” – should this be “more northern”?

Corrected to “more northern core range”

I. 91 – “where” – change to “wherein”.

Done.

Figure 1 – the pale, pastel colours of some species distributions don’t show up well in A, E and G.

Consider using the same colour for each species’ range in the separate figures.

Answer: We acknowledge that color perception can vary from one individual to another, which makes it difficult to reach a consensus on high-contrast colors that preserve the interest of the figure. The species range is not the central information on each panel and pastel colors are used to create a harmonious overlay with the background map. In turn, the dark colors have been retained for the findings on population genetic structure, which convey the main message of figure 1.

I. 149 – “similar to what Petit et al. found” – Can say “similar to Petit et al.” and specify for what species or what number of species.

Modified as “across several angiosperm trees”

I. 168 – change “reflects” to “better reflects”

Done.

I. 188 – “common mean estimates” – what does common mean here?

Answer: The sentence was modified as follows: “While the observed ratios were lower than usual estimates of N_e/N_c ”.

Figure 2 – it is fascinating how different the folded site frequency spectra are for the seven species. No change needed – just a comment. What is the yellow dot to the left of the species legend?

Answer: There should not be any yellow dots on the left of the species legend, we suppose that it comes from the conversion to another format. We will make sure that the figure looks as it should after file conversion.

I. 243 – change “tree species” to “these tree species”.

Done.

I. 272 – change “This situation” to “Amongst species included here, this” or something like that to make it clear you are referring to the current study.

Done.

I. 275 – “of their ranges, or have hybridized in the past”

Done.

I. 284 – change to “three clear species groups”

Done.

I. 316 – change “commonalities” to “some commonalities”

Done.

I. 339 – spell out names of genera.

Done.

I. 408-409 – There must be a typo here – “extreme values of heterozygosity ($\leq 6\%$ of the calls heterozygous). Should be 60% ?”

Answer: This is not a typo. We write:

“We also removed samples with excessive amounts of missing data ($\geq 60\%$ for all species except *B. pendula* where the threshold was $\geq 20\%$) or extreme values of heterozygosity ($\geq 6\%$ of the calls heterozygous) for *P. nigra* and *B. pendula*.”

In *Betula pendula* and *Populus nigra*, $\geq 6\%$ of calls across all SNPs being heterozygotes for a single individual was a clear outlier. Remember that the SFS is skewed towards rare alleles. Therefore, most SNPs are not heterozygous in a single individual. See below the graph we used to decide the threshold for e.g., *B. pendula*. Note that this figure is from the very initial quality check of the data. We suspect that these samples were interspecific hybrids and therefore wanted to exclude them from the final dataset.

I. 439-431 “To estimate the size of the genome with sufficient quality and depth for each species” – do you mean to estimate the size of the portion of the genome sequenced? Or the proportion of the genome sequenced in each species?

Answer: We meant the size of the portion of the genome sequenced with sufficient quality across individuals to call SNPs, for each species independently. We have modified the text to make it clear.

I. 453-454 – “To characterize the main genetic clusters among populations, we used SNP from dataset v.6.3.2 with only putatively neutral SNPs (i.e., 0-fold degenerate sites . . .)” = should be 4-fold consistent with I. 487-488?

Answer: The reviewer is correct, thanks for having spotted this typo that we have fixed.

Reviewer #2 (Remarks to the Author):

The authors compared the demographic history of seven widespread European tree species to infer the impact of multiple glacial cycles on their current genetic diversity using genomic data. To this end, they used site frequency spectra (SFS) to infer historical changes in effective population size (N_e) and tested alternative demographic scenarios using coalescent simulations. Despite the strong demographic impacts climate changes must have had on the studied species, there was no strong genomic signatures on N_e , which seems to have remained high, or has increased, during the Quaternary. Inferred N_e changes show time correlation between a few species but not over all of them, suggesting that species have responded in different ways to past environmental changes. In addition, evidence of past population fragmentation are typically much older than the last glacial maximum (LGM), showing that the genetic structure we see today cannot be interpreted considering solely the fragmentation-recolonization events that occurred over the last glacial cycle. The authors discuss the possible reasons that could explain the absence of a strong signal on N_e despite evidence of strong demographic changes in terms of census population sizes.

The originality of the study is that it compares seven tree species using a large amount of genomic data (sampling a few hundreds of individuals from 19 to 26 populations per species). In a sense, this study explores again the topic addressed in a famous article of Petit et al. (2003, cited by the authors) who compared the structure of 23 European species with chloroplast markers, using here many nuclear genes, which should provide more robust estimates (plastid genomes represent a single realization of stochastic evolutionary processes). Data analyses follow state-of-the-art procedures and I mostly agree with the conclusions of the authors.

Answer: We are glad that the reviewer appreciated the care put into the analyses and agrees with our general conclusions.

However, I have some major comments.

1) The work presented does not provide evidence that the data used are powerful enough to detect the cyclic demographic changes expected from glacial cycles. Therefore, I suggest to conduct coalescent simulations of a population going through repeated high and low N_e (e.g. 10 such cycles), adjusting the number of generations spent under high or low N_e to the typical duration of glacial and interglacial periods and to the range of generation times estimated for the seven tree species, considering different combinations of high and low N_e . Simulated SFS data using the ranges of sample sizes and numbers of

SNP's of the studied species could then be analysed using Stairway Plot 2 to confirm that, in principle, data are informative enough to detect cyclical changes of N_e . I wonder if a Stairway plot would detect demographic oscillations whenever N_e remains fairly high (say >1000 or 10000) during the low N_e phases, even if the N_e during high N_e phases is several orders of magnitude larger.

Answer: First, we would like to reiterate that our manuscript's main purpose was not to identify all demographic events that have affected species' census population size. Rather, we wanted to make the point that such events may not have been long and drastic enough to leave major signatures in genetic diversity of species with generally large N_e , outcrossing mating systems, extensive gene flow and long, overlapping generations. From that perspective, what appears as a lack of statistical power in the data also indicates that the effects of glacial cycles on genetic diversity are less important than often assumed, at least for this type of species.

We have now further emphasized this in the Discussion section:

“Very large and genetically connected populations along with long generation time—hence, a limited number of generations with elevated drift—allowed forest tree species to retain genetic diversity through time, despite intermittent, substantial geographical range contractions.”

“On the evolutionary time scale, glacial cycles are shorter and recur faster for species with longer generation time and larger N_e .”

Comprehensive analyses of the statistical power and properties of demographic inference methods are outside of the scope of the manuscript and partly addressed, e.g., in Liu and Fu (2015) and (2020) on the Stairway Plot method. We have carried out simulations as suggested by the reviewer but omit including them into our results to avoid losing the focus of the manuscript.

We agree with the reviewer that if N_e remains high during glacial periods, it may be difficult to recover the oscillations. Other aspects include the intensity of the demographic events, the mutation rate, the number of samples and the size of the sampled genome. This is a vast space of parameters to explore, and we are not addressing all aspects here.

We tested the ability of Stairway Plot 2 (SP2) to recover demographic cycles for several values of generation time (1, 15, 25 and 60 years per generation), size of the sampled genome (1 Mbp or 6 Mbp), different mutation rates (2.7×10^{-8} or 7.7×10^{-9} per site per generation), and current effective population sizes (N_e ; 10K, 100K, 1M; the latest demographic event being an expansion, this is the highest N_e). The intensity of the expansion / contraction of population size was set to 10 fold, and we implemented 10 events (i.e., 5 cycles of expansion followed by a contraction), following the expected times of the glacial / interglacial periods: a demographic expansion 15 Kya, a decline 120 Kya, then following a period of 120 Kya, expansion or decline events successively, until a final decline about 1 Mya. Each event was modeled as an instantaneous change of N_e with stable N_e in-between. These models were simulated with Fastsimcoal2 to produce folded site frequency spectra (fSFS) and using a sample size of 20 haploid genomes. We then ran demographic analyses of the fSFS with SP2 using the settings used in our study (4 breakpoints fixed at sample sizes following manual instructions and 200 simulations). Find below the

plots summarizing the results with, in blue lines, the expected model and in black the SP2 inference. We also report nucleotidic diversity (π) of each fSFS generated.

Across most analyses, SP2 does not capture the full sets of demographic events implemented in the models. At most, the analyses correctly infer 2 out of 10 demographic events approximately at the correct time. In most cases, the inference only recovers the most recent demographic event. Across the parameter values we explored, this seems to be true notably if the starting (present time) N_e was low (10K), and / or when the generation time was short. When N_e was high (100K or 1M), SP2 only recovered the most recent expansion if the generation time was short (1 if $N_e = 1M$, or at most 25 if $N_e = 100K$). When both N_e and generation time were high, SP2 instead inferred old collapses of populations (older than 100K years ago). Overall, these results are consistent with Liu and Fu (2015). Indeed, Liu and Fu noted that the Stairway plot approach was more sensitive to recent demographic events and that independent changes of population sizes occurring in a short span of time are difficult to detect. In the cases where we only detect old collapses, this is likely due to the fact that when N_e is high (even during glacial periods) and the populations have such a long generation time, the short interglacial periods do not allow recovery of genetic diversity. This is consistent with the reviewer's prediction.

It is noteworthy that

1. Only a fraction of our simulations result in the nucleotide diversity observed in our real data (magnitude of 10^{-3}). We typically reach that only with a population size of 1M. Consequently, as the oscillations are scaled with N_e , they become so fast and frequent that it is understandable that there is no strong signal of them.
2. None of the simulated datasets reproduce the pattern we observe in our Stairway plot analyses. This indicates that the true changes in N_e are not captured by the simple oscillation simulations conducted here. It is impossible to know the true oscillation pattern, and / or find it by iterating the large number of parameters involved.

As a conclusion, we agree that current data and methods cannot recover all past changes in N_e in a scenario as complex as has likely occurred during the past climatic oscillations. However, we do not claim to have this power in the manuscript. Our goal was to emphasize how tree species' biological and life-history traits make them genetically resilient to such oscillations, and to some extent, our simulations support this interpretation.

Fig. 1. Stairway plot 2 inference of the change of effective population size (N_e) over time (in years, from present to past) of an oscillating population. The black lines represent the median estimates (over 200 simulations), dark and light shades are respectively the 95% and 99% confidence intervals. The blue and dashed line represent the theoretical model simulated with Fastsimcoal2. Each panel corresponds to a different starting N_e and generation time (Gt). Across simulations, 20 haploid genomes were simulated with a sample size of the genome of 1.5 Mbp (15K contigs of length 100 bp) and a mutation rate of 7.7×10^{-9} per site per generation.

Fig. 2. Same as in Fig. 1 but simulations with sample size (S_s) of 20 haploid genome, with genome size (G_s) size (G_s) of 6 Mbp (60K contigs of length 100 bp) and a mutation rate of 7.7×10^{-9} per site per generation.

Fig. 3. Same as in Fig. 1 but simulations with sample size (Ss) of 20 haploid genomes, with genome size (Gs) of 1.5 Mbp (15K contigs of length 100 bp) and a mutation rate of 2.7×10^{-8} per site per generation.

Fig. 4. Same as in Fig. 1 but simulations with sample size (S_s) of 20 haploid genomes, with genome size (G_s) of 6 Mbp (60K contigs of length 100 bp) and a mutation rate of 2.7×10^{-8} per site per generation.

2) I'm concerned with the interpretation of SFS data in terms of past N_e . Fundamentally, the method used to infer past N_e from SFS consists in estimating how rates of coalescence vary through time, and this can indeed be interpreted in terms of N_e (basically the inverse of coalescence rate) assuming an isolated and panmictic population. The problem is that the studied species do not form a single isolated panmictic population but are structured in multiple populations. Most of the recent literature on demographic inference based on SFS makes the approximation that population structure can be ignored and directly convert estimated coalescence rates into N_e but this is problematic, as shown for example by Mazet et al. (2016), Chickhi et al. (2018) or Arrondondo et al. (2021). The current literature reminds me the situation three decades ago when many articles published estimates of the number of migrants per population (N_m) based on F_{st} , until a famous article by Whitlock & McCauley (1999) recalled that the conditions under which $F_{st} = 1/(1+4N_m)$ are very restrictive (island model at drift-migration equilibrium and low mutation rate) and probably not met in most real conditions. I think the scientific literature today is a bit in the same situation with SFS and past N_e estimates. Therefore, I believe the present ms could do a better job by controlling the impact of population structure. To this end, first I suggest to interpret the Stairway plots in terms of IICR (inverse instantaneous coalescence rate) rather than in terms of past N_e , as recommended by Chickhi et al. (2018). Second, additional coalescent simulations of a structured population would help assess whether population structure itself is not contributing to a certain stability of the IICR across glacial cycles. This could be done by simulating over multiple glacial cycles a system of 8 populations: 4 permanent populations corresponding to refuge areas (which might be larger during glacial than interglacials) would each be connected to one of 4 unstable populations that are very large during interglacials and small (or vanish) during glacials. Gene flow would occur in a stepping-stone manner only between the unstable populations during interglacials, and between each of the latter populations with its source (refugial) population. The four isolated refugial populations might generate the population structure observed in the majority of studied species (four genetic clusters). The resulting SFS and IICR estimates from Stairway plots could then be compared with the one obtained for a single population undergoing similar overall demographic changes. I believe such simulations could help interpret the data and may help explain the stability of stability of the IICR across glacial cycles.

Answer: The IICR is defined for two alleles and is “equivalent to the past temporal trajectory of N_e previously defined as the coalescent N_e (Sjödín et al. 2005) in a panmictic population under neutrality and it is the quantity estimated by the popular PSMC method of Li and Durbin (2011)”.

We agree that the IICR is a better technical name for the inference made by Stairway plot 2. However, we argue that this level of nuance is useful only for population geneticists, and will only confuse the broader audience. Therefore we prefer to use the more intuitive terminology of effective population size while explaining in the text the caveat of the interpretation.

While the methods developed by Chikhi et al. (2010, 2018), Mazet et al. (2016), Arrondondo (2021), or Mazet (2023) are all valuable and interesting steps towards a better demographic inference that could account for population structure, we note that, so far, these tools do not yet fully correct the effects of genetic structure on demographic inferences, or only do so in a limited manner. For example, Mazet et al. 2023 (PCI vol 3, 2023, page 7) say “*We then showed that in the simple case of a population structured in islands, then adding the information of the T_3 distribution to the T_2 distribution is enough to distinguish this model from the panmictic model having the same T_2 distribution, thus the same IICR (Grusea et al., 2018). This result provides theoretical evidence that a sample size strictly greater than two is sufficient to distinguish*

an island structured model from a panmictic model, but initial attempts to move into practice have not yet been successful, because of the precision required, which often blends into the noise of the real data”.

Or, (Mazet et al. 2023, pages 11-13):

“Nevertheless, the average SFS is theoretically known only for a panmictic model (see for example Griffiths and Tavaré (1998)), but the calculation becomes of great combinatorial complexity for any structured model. In the case of the island model, Armando Arredondo has just completed the theoretical treatment of obtaining the average SFS for any value of k , as well as the feasibility in computation time for a sample size of $k \leq 26$ in the current state of computing capabilities (Arredondo Soto, 2021, chapter 3). It now remains to implement this algorithm in the inference software.

[...]

The practical problem comes from the fact that the larger the sample size, the shorter the coalescence time, and thus the fewer the genomic traces on the data, because the number of mutation and recombination events decreases very quickly, and falls below the acceptable threshold for the statistical estimation to be satisfactory.”

However, we fully agree with the reviewer that population structure is an important issue that needs consideration during demographic inferences. In fact, this is something that we have cautiously considered in our analyses and that led us to base our main interpretation on the one-sample-per-population sets, instead of the full sets of samples. The approach we have adopted is based on the reasoning that subsampled data will allow us to focus our inference on the “collecting” phase of the genealogy, past the effect of structure (Wakeley et al. 2001). This reduces the issue of structure (at the cost of statistical power). Further, we believe we are in a situation where the issue should be relatively limited compared to other organisms. Indeed, all species analyzed are wind-pollinated trees, with high migration rate. Our analyses of structure also support low levels of population structure, even at the continental scale.

We have now clarified that different subsamples were used for inference to account also for effects of population structure:

“Importantly, to account for the effect of sampling and population structure on demographic inferences²⁹, we conducted analyses at the species, population, and one-sample-per-population levels.”

and added the citation to Chikhi et al. 2010.

The suggestion to carry out extensive simulations of structured populations makes sense on the first sight, but in practice is hard to execute in a way that would be relevant for the seven species where population structure can be extremely weak (*Pinus sylvestris*) to more pronounced (*Pinus pinaster*) and at the same time for a range of effective population sizes, generation times, and life-history traits that would cover the biological models we are investigating. Furthermore, we have very limited information on the number of putative refugia beyond the last glacial maximum (LGM), and the available paleoecological data suggest that, for some species, such refugia might simply have not existed, the species surviving in a set of populations scattered south of the glaciated areas. Thus, we chose not to incorporate simulations into the manuscript because, reporting the results of such extensive simulations and methodological evaluation would significantly change the focus of the manuscript, which we wish to avoid, and is hard to implement in a way the reviewer is suggesting.

However, to assess the impact that population genetic structure may have in our analyses, we ran three demographic inferences with Stairway plot for each of the seven species: We first pooled the two most diverged populations together (“mixed”), and then also analyzed each population separately. The results are reported below. We can see that in the analysis of the mixed populations, Stairway plot infers demographic events that are not seen in either population individually in two of the seven species: *Fagus sylvatica* and *Quercus petraea*, but these differences may be artifactual. In all other cases, the inference is consistent across the mixed and separate analyses, suggesting that the level of population genetic structure we have in our data has, even in the worst possible cases (where we pool entire populations from the most extreme parts of the distribution), limited effects.

Fig. 5. Stairway plot inferences of change in effective population size (N_e) over years (from present to past) and conducted, for each species, on two populations mixed together (leftmost panels), or on each population separately (middle and rightmost panels). From top to bottom, we show the results for *Picea abies*, *Betula pendula*, *Fagus sylvatica*, *Populus nigra*, *Pinus pinaster*, *Pinus sylvestris* and *Quercus petraea*. The populations presented are the same as those used in the manuscript to represent the northern and southern genetic pools (divergence analysis).

3) Table 1 and p. 13: the reciprocal of the slope of pairwise $F_{st}/(1-F_{st})$ against $\log(\text{distance})$ is interpreted as a neighbourhood size ($N_s = 4 * \pi * D * \sigma^2$). This is correct at drift-dispersal equilibrium but given the

large spatial scale of the sampling, the detection of different genetic clusters and the history of populations subject to glacial cycles, drift-dispersal equilibrium is probably not reached at the studied scale. Nevertheless, the slope of pairwise $F_{st}/(1-F_{st})$ against $\log(\text{distance})$ remains a useful synthetic measure of the intensity of the isolation-by-distance pattern. Hence, I suggest reporting these slopes instead of biased estimates of N_s .

Answer: The reviewer is correct about the assumption made when computing the Neighborhood size (N_s) from the regression of pairwise $F_{st}/(1-F_{st})$ against $\log(\text{distance})$ and we did not state this plainly in the main text. As N_s was only provided as a way to compare IBD patterns across species and not for its value per se, we followed the reviewer's advice and directly reported the slope of the regression in the table. The main text and the caption of the table have been adjusted accordingly.

4) Line 492. "To investigate the change in population differentiation along latitude, longitude and elevation, we computed the average population-specific F_{ST} scaled by geographic distance. For each focal population, we computed the average pairwise F_{ST} divided by the average distance (km) between each pair of populations in which the focal population was involved." The proposed scaling would be justified if pairwise F_{st} 's tended to increase linearly with the spatial distance. However, as shown in Fig 1, pairwise F_{st} 's tend to increase more or less linearly with the $\log(\text{spatial distance})$. Hence, I think F_{st} scaling should use $\log(\text{distance})$ instead of distance.

Answer: Thank you for the suggestion. The recommendation made by the reviewer did improve the fit of the linear regressions, but it did not change the main trends in the result. We thus modified the text in material and methods and changed the corresponding figure (S2), but we left the result section of the main text untouched.

Minor comments:
Line 247: "Very large and genetically connected populations allowed tree species to retain genetic diversity through time, despite intermittent, substantial geographical range contractions." The long generation time of trees, implying a limited number of generations of higher drift when populations are small, is probably an important feature limiting the loss of genetic diversity.

Answer: The reviewer is correct; we have modified the sentence as follows: "This potential is likely a reflection of their unique biological features. Very large and genetically connected populations along with long generation time - hence, a limited number of generations of higher drift - allowed tree species to retain genetic diversity through time, despite intermittent, substantial geographical range contractions."

Fig. S26: Inset figures are too small to interpret. I suggest to present them in 2 or 3 pages.

Answer: We agree. As suggested, we split the figure in 2 pages to improve clarity.

Table S9: duration seems to be expressed in centuries rather than in Kya.

Answer: We checked and it is actually Kya.

Table S17: "the portion of the reference genome in which at least 50% of the samples of the given species have a minimum coverage of 8x and a minimum genotype quality value of 20." Does this refers to the last column ("Available genome") and is it expressed in bp?

Answer: The reviewer is correct, we have modified the caption of the table to make it clear.

Cited references:

- Arredondo, A., Mourato, B., Nguyen, K., Boitard, S., Rodríguez, W., Noûs, C., Mazet, O., & Chikhi, L. (2021). Inferring number of populations and changes in connectivity under the n-island model. *Heredity*, 126(6), 896–912. <https://doi.org/10.1038/s41437-021-00426-9>
- Chen J, Li L, Milesi P et al. (2019). Genomic data provide new insights on the demographic history and the extent of recent material transfers in Norway spruce. *Evolutionary Applications* 12, 1539-1551.
- Chikhi, L., Sousa, V. C., Luisi, P., Goossens, B., & Beaumont, M. A. (2010). The Confounding Effects of Population Structure, Genetic Diversity and the Sampling Scheme on the Detection and Quantification of Population Size Changes. *Genetics*, 186(3), 983–995. <https://doi.org/10.1534/genetics.110.118661>
- Chikhi, L., Rodríguez, W., Grusea, S., Santos, P., Boitard, S., & Mazet, O. (2018). The IICR (inverse instantaneous coalescence rate) as a summary of genomic diversity: Insights into demographic inference and model choice. *Heredity*, 120(1), 13–24. <https://doi.org/10.1038/s41437-017-0005-6>
- Hanlon VCT, Otto SP, Aitken SN (2019). Somatic mutations substantially increase the per-generation mutation rate in the conifer *Picea sitchensis*. *Evolution Letters* 3, 348-358.
- Liu X, Fu Y-X (2015). Exploring population size changes using SNP frequency spectra. *Nature genetics* 47, 555-559.
- Liu X, Fu Y-X (2020). Stairway Plot 2: demographic history inference with folded SNP frequency spectra. *Genome Biology* 21,
- Mazet, O., Rodríguez, W., Grusea, S., Boitard, S., & Chikhi, L. (2016). On the importance of being structured: Instantaneous coalescence rates and human evolution--lessons for ancestral population size inference? *Heredity*, 116(4), 362–371. <https://doi.org/10.1038/hdy.2015.104>
- Mazet, O., & Noûs, C. (2023). Population genetics: Coalescence rate and demographic parameters inference. *Peer Community Journal*, 3, e53. <https://doi.org/10.24072/pcjournal.285>
- Wakeley, J., & Aliacar, N. (2001). Gene Genealogies in a Metapopulation. *Genetics*, 159(2), 893–905.
- Willyard A, Syring J, Gernandt DS, Liston A, Cronn R (2007). Fossil calibration of molecular divergence infers a moderate mutation rate and recent radiations for *Pinus*. *Molecular Biology and Evolution* 24, 90-101.

REVIEWERS' COMMENTS

Reviewer #1 (Remarks to the Author):

I am fully satisfied with the authors' responses to my comments and suggestions through two rounds of review and editing, and support the current version of the manuscript being accepted for publication in Nature Communications.

Reviewer #2 (Remarks to the Author):

Globally I appreciate the way the authors have addressed most of my comments. Notably, they performed new data analyses on simulated populations to better assess the behaviour of Stairway Plot inferences, which illustrate well the potential signatures that repeated demographic changes can leave on estimates of past coalescence rates. Although the authors did not judge useful to include these results in the new version, I personally think that they help interpret the results and would merit their inclusion as supplementary material (see suggestions below). In particular, these simulations highlight that under high fluctuating population sizes (with 10-fold cyclic changes), long generation time and low mutation rate, we should not expect to detect a signal of cyclic demographic changes in Stairway Plots with the type of data available. The signal recovered is then a smooth decline of inferred N_e , while the analyses on real data reveal a trend of increasing N_e trajectory.

The authors do not wish to reinterpret their N_e estimates in terms of IICR because readers might be lost by these considerations. I respect this choice but I think it is then important to inform readers of the limits of SFS-based N_e inference because the current ms might give the impression that the methods used reliably infer past changes in population sizes, in contrast to what simulations show. Line 182, I suggest to replace "Importantly," by a text conveying the following idea: "SFS-based N_e trajectory estimations measure changes in coalescence rates of gene genealogies, which depend on historical changes in N_c affecting instantaneous N_e but also on barriers to gene flow in structured populations and the way gene genealogies are sampled [REF]. To account for the effect of".

Line 206, I suggest to add the following idea: "Simulations of unstructured populations undergoing cyclic 10-fold demographic changes confirm that N_c fluctuations do not translate in fluctuating SFS-based N_e trajectories when instantaneous N_e remains large ($>100,000$) and generation time is long (≥ 15 years); the inferred N_e trajectories rather suggest an ancient decay of N_e (supplem. material X)."

Line 265, I suggest adding "Simulations of cyclic demographic changes following glacial cycles illustrate that while the inferred N_e trajectory of annuals can reflect the demographic fluctuations of the last glacial-interglacial cycle, this is not the case of organisms with much longer generation time like trees (supplem. material X)."

Finally, lines 285-318, the authors do not attempt to explain why most species show a trend of increasing N_e trajectory on the long-term (rather than a constant N_e). I think this could result from the choice of the species: the seven studied tree species are not a random selection of the current European tree flora but are relatively abundant species. Hence, they represent

ecologically successful species, at least in the late Quaternary. However, for the little I know, before the Quaternary, the European Pliocene tree flora was quite different and more diverse, including many species now extinct. The increasing N_e trajectory of the studied species may thus reflect that the authors choose a set of particularly successful species during the Quaternary. Whether rarer European tree species show similar N_e trajectories remain to be investigated. I suggest the authors elaborate on these ideas.

RESPONSE TO REVIEWERS' COMMENTS

find below our point-by-point answers to remaining comments from the reviewers. Our answers are in blue font.

Reviewer #1:

I am fully satisfied with the authors' responses to my comments and suggestions through two rounds of review and editing, and support the current version of the manuscript being accepted for publication in Nature Communications.

Reply: We thank the reviewer for their appreciation of our work.

Reviewer #2:

Globally I appreciate the way the authors have addressed most of my comments. Notably, they performed new data analyses on simulated populations to better assess the behaviour of Stairway Plot inferences, which illustrate well the potential signatures that repeated demographic changes can leave on estimates of past coalescence rates. Although the authors did not judge useful to include these results in the new version, I personally think that they help interpret the results and would merit their inclusion as supplementary material (see suggestions below). In particular, these simulations highlight that under high fluctuating population sizes (with 10-fold cyclic changes), long generation time and low mutation rate, we should not expect to detect a signal of cyclic demographic changes in Stairway Plots with the type of data available. The signal recovered is then a smooth decline of inferred N_e , while the analyses on real data reveal a trend of increasing N_e trajectory.

Reply: We thank the reviewer for their positive comments. We have considered all the comments below and notably included the simulation work as Fig. S28 to S31 and S33 in supporting information, as well as a description of the methods used and the main results in the main document.

The authors do not wish to reinterpret their N_e estimates in terms of IICR because readers might be lost by these considerations. I respect this choice but I think it is then important to inform readers of the limits of SFS-based N_e inference because the current ms might give the impression that the methods used reliably infer past changes in population sizes, in contrast to what simulations show.

Line 182, I suggest to replace "Importantly," by a text conveying the following idea: "SFS-based N_e trajectory estimations measure changes in coalescence rates of gene genealogies, which depend on historical changes in N_c affecting instantaneous N_e but also on barriers to gene flow in structured populations and the way gene genealogies are sampled [REF]. To account for the effect of".

Reply: We agree with the reviewer and have modified the text as suggested: "SFS-based inference of N_e trajectory measure changes in coalescence rates of gene genealogies, which depend on historical changes in N_c affecting N_e but also on barriers to gene flow in structured populations and the way gene genealogies are sampled (Mazet and Noûs 2023 and references therein)."

Line 206, I suggest to add the following idea: “Simulations of unstructured populations undergoing cyclic 10-fold demographic changes confirm that N_c fluctuations do not translate in fluctuating SFS-based N_e trajectories when instantaneous N_e remains large ($>100,000$) and generation time is long (≥ 15 years); the inferred N_e trajectories rather suggest an ancient decay of N_e (supplem. material X).”

Reply: Following the reviewer’s suggestion we have incorporated the following text: “Simulations of unstructured populations undergoing cyclic 10-fold demographic changes confirm that N_c fluctuations do not translate in fluctuating SFS-based N_e trajectories when N_e remains large ($>100,000$) and generation time is long (≥ 15 years) but rather suggest an ancient decay of N_e (supplementary material Fig. S28 to S31).”

Line 265, I suggest adding “Simulations of cyclic demographic changes following glacial cycles illustrate that while the inferred N_e trajectory of annuals can reflect the demographic fluctuations of the last glacial-interglacial cycle, this is not the case of organisms with much longer generation time like trees (supplem. material X).”

Reply: Thanks. We have added the following sentence: “Simulations of demographic changes following glacial cycles show that inferred N_e trajectory of annuals can capture the demographic fluctuations of the last glacial-interglacial cycle. However it is not the case for organisms with much longer generation time like trees (supplementary material Fig. S28 to S31).”

Finally, lines 285-318, the authors do not attempt to explain why most species show a trend of increasing N_e trajectory on the long-term (rather than a constant N_e). I think this could result from the choice of the species: the seven studied tree species are not a random selection of the current European tree flora but are relatively abundant species. Hence, they represent ecologically successful species, at least in the late Quaternary. However, for the little I know, before the Quaternary, the European Pliocene tree flora was quite different and more diverse, including many species now extinct. The increasing N_e trajectory of the studied species may thus reflect that the authors choose a set of particularly successful species during the Quaternary. Whether rarer European tree species show similar N_e trajectories remain to be investigated. I suggest the authors elaborate on these ideas.

Reply: We thank the reviewer for this suggestion to which we agree. We have added the following text and references:

It is worth pointing out that the seven species considered in this study are all widely distributed, relatively abundant, and ecologically important species of European forests. As such, they are interesting and important models to study the effects of environmental factors on the evolutions of European forests across space and time. However, they do not form a representative sample of the modern European tree flora. Furthermore current species are those that survived past mass extinctions. At the Pliocene-Pleistocene transition the climate in the northern hemisphere changed dramatically with the onset of the glacial-interglacial periods, resulting in large scale extinction of trees, especially in Europe (Rull, 2020). The modern flora represents less than 30% of the tree genera present during the Tertiary (Eiserhardt et al, 2015; Latham and Ricklefs, 1993). Extinctions eliminated deterministically

cold sensitive species (Svenning, 2003), and following episodes of selection further favored species sharing invasiveness attributes (prolificity, competitive ability, dispersal; Lamarque et al, 2011) that facilitated locally the replacement of the extinct species.

References:

Eiserhardt WL, Borchsenius F, Plum CM, Ordonez A, Svenning JC (2015). Climate-driven extinctions shape the phylogenetic structure of temperate tree floras. *Ecology Letters* 18(3): 263-272.

Lamarque JL, Delzon S, Lortie CJ (2011). Tree invasions: a comparative test of the dominant hypotheses and functional traits. *Biological Invasions* 13: 1969-1989.

Latham RE, Ricklefs RE (1993). Global patterns of tree species richness in moist forests. Energy-diversity theory does not account for variation in species richness *Oikos* 67(2): 325-333.

Rull V (2020). *Quaternary ecology, evolution and biogeographic*. Academic Press.

Svenning JC (2003). Deterministic Plio-Pleistocene extinctions in the European cool-temperate tree flora. *Ecology Letters* 6(7): 646-653.